# Relationship between the surface and internal deformation features of toppling in anti-dip rock slopes

Guang Zheng, Mingyu Zhang⊚*, Minghao Chen, Qi Liu, Jinning Yang, Jun Chen, Huiyan Lu

State Key Laboratory of Geohazard Prevention and Geoenvironment Protection, Chengdu University of Technology, Chengdu, Sichuan, China

* zhangmingyu1@stu.cdut.edu.cn

## Abstract

Multiple researchers have effectively utilized InSAR for early identification research in southwest China. However, the intricate geological structure in this region includes deeply buried and unpredictable anti-dip stratiform rock slopes, posing an additional challenge for InSAR remote sensing. This study establishes a coupling relationship between slope topple deformation patterns and surface macro-deformation characteristics through a centrifuge model test, analyzing slope bending and topple deformation stages. The InSAR interpretation of prototype slopes is employed to retrospectively infer the slope stage and visualize the slope deformation process. This research provides technical assistance for identifying and preventing slope failures. The result demonstrates that: (1) The failure process of anti-dip stratiform rocky slope involves four major stages. Failures often begin at the slope's base and move to the middle and top regions, eventually resulting in entire slope failure when the stepped fracture surface forms. (2) Experimental image recognition analysis was conducted to monitor the rate changes in the deep and surface layers, establishing a corresponding coupling relationship between them. Observations reveal that significant deformation areas exhibit a bottom-to-top development pattern. (3) A comparison of the InSAR interpretation results for the Zhayong deformation with the test conclusions reveals that the landslide was in the progressive deformation stage. In summary, this study provides valuable technical support for utilizing InSAR technology to identify and prevent slope failures in complex geological conditions.

## 1. Introduction

Landslides, the most prevalent geological hazard, are widespread, frequently occurring, and pose significant threats to human life and property [1]. Traditional research on slope catastrophe processes, monitoring, and early warning methods focuses mainly on the establishment of landslide monitoring networks to capture the changing patterns of specific indicators [2]. However, these traditional methods face significant limitations in vegetated or steep areas at

**Data Availability Statement:** All relevant data are within the manuscript and its Supporting Information files.

**Funding:** This study is funded by the National Key R&D Program of China (No. 2021YFC3000401).

**Competing interests:** The authors have declared that no competing interests exist.

the crest of slopes. This opens up opportunities for InSAR remote sensing technology to monitor the macroscopic deformation characteristics associated with geological hazards, as several academics are currently using InSAR for early detection [3]. This technology surpasses conventional surface deformation monitoring methods, enabling dynamic tracking and analysis of typical surface deformation features, including background subsidence and local deformation. However, these macroscopic surface deformation patterns may not be adequate for InSAR to determine the destructive processes of slope instability and the stage at which they occur.

Toppling is a fundamental type of slope movement commonly observed in rock slopes with regular foliation layers [4–7]. Since the mid-20th century, increasing human engineering activities have revealed numerous occurrences of rock toppling in anti-dip stratiform rocky slopes [8]. Goodman classified toppling into three main types: flexural, block, and block-flexure toppling, based on distinct failure mechanisms [9]. Nichol et al. classified toppling into ductile and brittle failure behaviors based on lithology [10]. Subsequently, many researchers have investigated the formation and development mechanisms of toppling in anaclinal rock slopes [11–14]. Zhang et al. compiled 21 impact factors for landslide damage, used the discrete element code to obtain the toppling response of the structure, determined the order of the impact factors based on their influence weights, and found that the slope failure surfaces were controlled by cross joints, and that rocks in the slope toe played an inhibiting role in preventing slope deformation [15].

Physical model tests are commonly employed to investigate slope deformation, providing direct observation of damage, deformation, and the progression of slope movements. Aydan et al. utilized base friction models to study flexural toppling and block toppling phenomena [16]. Amini et al. employed a novel tilting table to simulate block bending toppling, and secondary toppling, and proposed a corresponding theoretical framework [17]. Adhikary et al. conducted a series of centrifuge experiments, discovering that two failure modes (instantaneous and progressive failure) were influenced by the joint friction angle [18]. Dong et al. investigated the impact of anchorage angles on toppling deformation through laboratory-scale tests [19]. Zhang et al. developed a physical model of an anaclinal rock slope that simulates river down cutting, using the Linda landslide as a site case physical model, and classified the deformation stages of the slope and the deformation zones of the rock mass based on multi-field monitoring [20]. Fang et al. explored the effectiveness of multi-field monitoring in physical modelling tests, established a slope model under the action of excavation, used the inverse velocity method for damage prediction of the physical model, and investigated the deformation characteristics and damage mechanism of the slope through multi-field monitoring [21].

In general, centrifuge and full-scale experiments are more advanced than 1 g indoor physical model tests in describing landslide damage processes; the individual effects of different parameters on landslides can be investigated by using simplified boundaries and geometries in an artificial high gravity field; and the gradual process of geohazards from the beginning to the end of the process has a long time course and is difficult to observe comprehensively, but full-time, full-process observations can be carried out more easily by centrifugal model experiments. Centrifugal modelling experiments make it easier to observe the entire process over the full duration [22]. Adhikary et al. utilized materials like glass, gypsum, and fiberboard to create a model of an anti-dip rocky slope and simulated transient toppling damage and progressive toppling damage through testing [23]. Stewart et al. simulated toppling damage during excavation using centrifugal tests [24]. Tobita, T et al. investigated the generation of lateral displacements in the lengthwise direction of the model surface caused by slope surface bending in various radial gravity fields [25]. Jin et al. used a large centrifugal system to study the evolution of block tipping, which showed that block tipping is mainly caused by bending damage rather

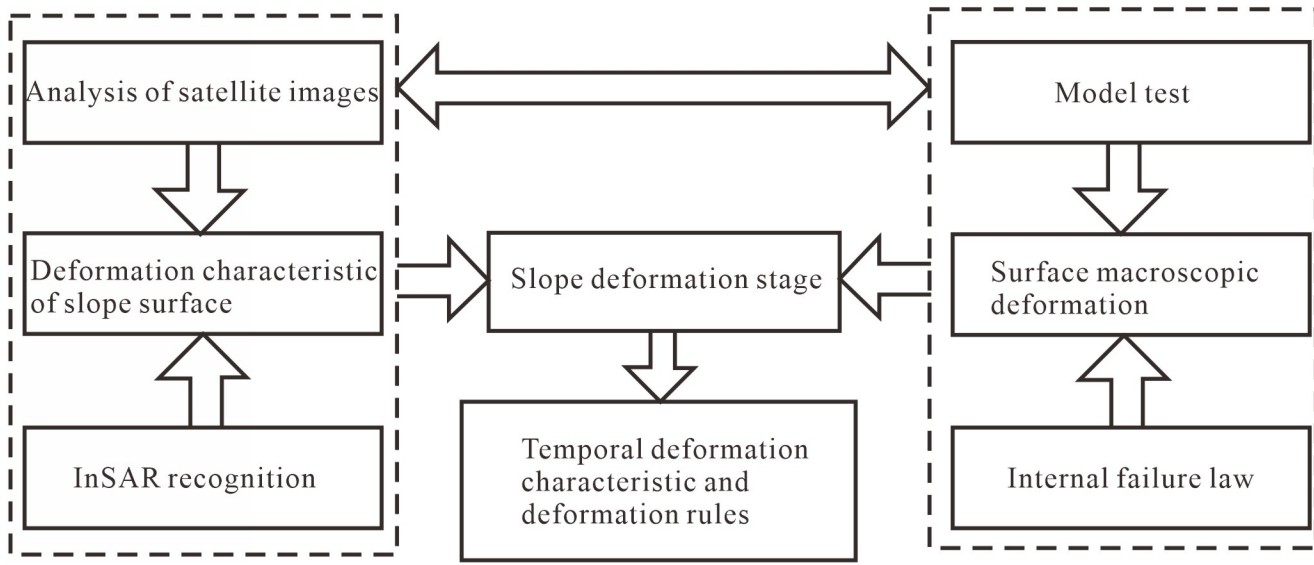

**Fig 1. The idea of the study.**

than shear damage, and that tensile strength is also a key factor in the evolution of block toppling [26]. Jia et al. simulated the deformation and damage processes, damage modes, and damage paths of the slope under rainfall and excavation conditions using centrifuge technology to study the destabilization mechanism of anticlinal layered soft metamorphic rock landslides [27].

Although previous researchers have conducted in-depth studies on the deformation evolution mode of toppling-fracture, there are still gaps in understanding the corresponding relationship between the macroscopic deformation characteristics of the toppled deformed body and the internal rupture.

In this study, building upon previous research on monitoring and warning models, we conduct centrifugal model test research on anti-dip stratiform rocky slopes. During the tests, we monitor the deformation of the deep rock slabs and the macro deformation characteristics of the ground surface, divide the stages of rock toppling deformation, investigate the macro deformation characteristics of the ground surface corresponding to each stage, establish the corresponding coupling relationship between the toppling deformation pattern of the slope and the macro deformation characteristics, guide the deformation results of the actual slope's InSAR interpretation and the field investigation, summarize the deformation characteristics and rules, and visualize the deformation stages, This allows us to improve the understanding of the deformation characteristics and internal rupture of the slope. Visualizing the deformation stages of slopes improves the accuracy of early identification and provides technical support for the identification, prevention, and control of major landslide disasters. The specific research idea is shown in Fig 1.

## 2. Geological background of the test prototype slope

### 2.1 Terrain topography

The Zha Yong toppling deformation is located approximately 25 km from Polo Village and 5 km from Jinsha Village, at coordinates 31°16′36.42″N and 98°44′22.29″E. The relative elevation difference from the Jinsha River surface is about 1375 m, and the peak elevation of the

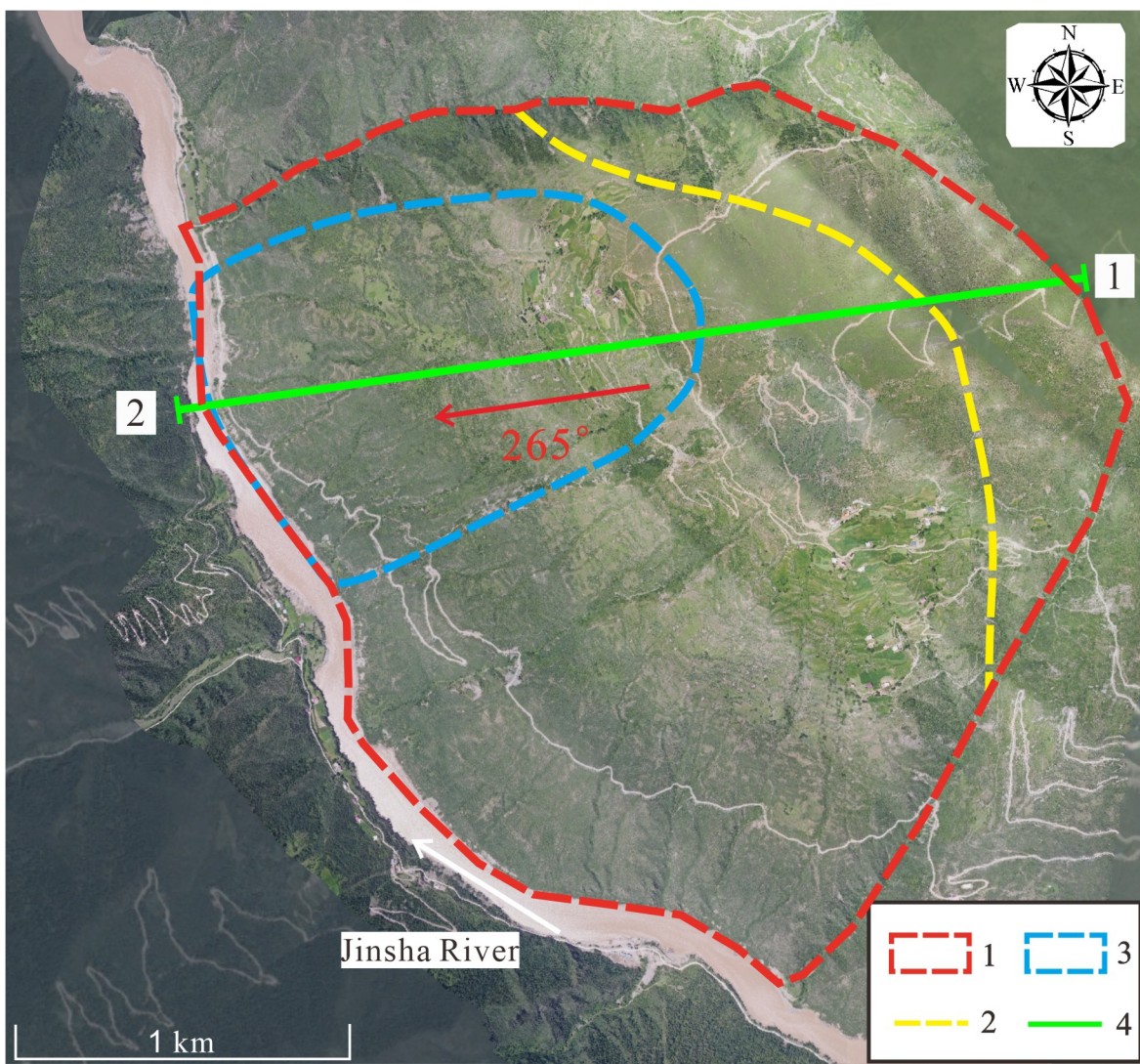

**Fig 2. ZhaYong deformation panorama. 1.** Deformation boundary range **2.** Steep and gentle junction line **3.** Major deformation zone **4.** 1–2 Geological profile line.

slope is approximately 4275 m. The two sides of the deformed area exhibit no significant topographic relief variation. The Zha Yong deformation measures 2949 m in length and 2469 m in width. The landscape features a large, planation-surfaced mountain characterized by steep topography, deep river cuts, and sparse vegetation. The deformation exhibits a bell-shaped profile, with a narrow top that broadens towards the base and extends outward at the toe, as shown in Fig 2.

## 2.2 Structural characteristics of the deformation zone

The toe of the Zha Yong toppling deformation extends along the river. The slope orientation is approximately NW20˚, with dip direction of about SW70˚. Exposed bedrock near the downstream side wash belongs to the T3l formation of the Triassic Lanashan Group. The lithology consists primarily of phyllite with a minor proportion of medium- to thin-grained sandstone. The rock types are predominantly Chlorite Sericite Phyllite, arkose, and calcareous Siltstone,

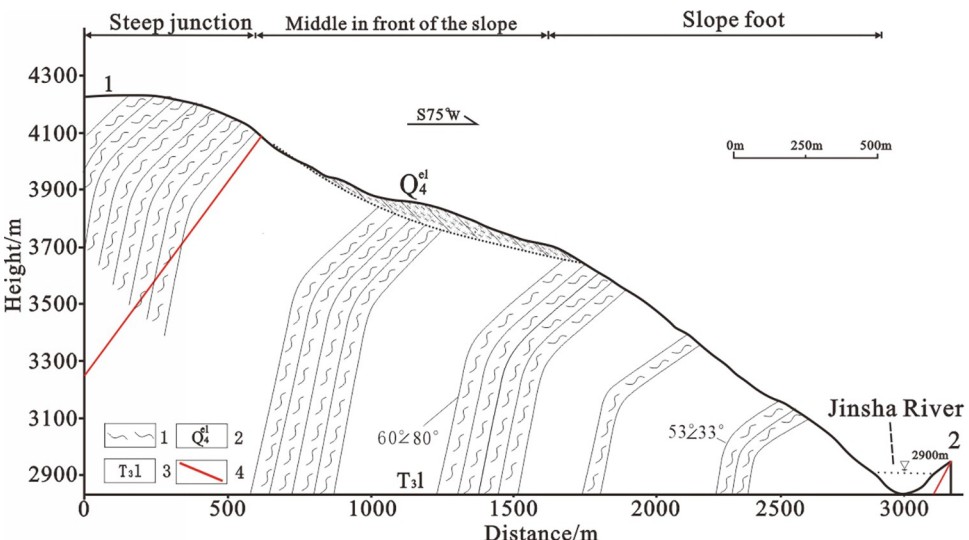

**Fig 3. Geological profile of the ZhaYong toppling deformation. 1.** phyllite **2.** Q4edl **3.** Triassic Rana Mountain Formation **4.** Mount Ella fault branch.

with millipede being the most abundant. The original rock structure exhibits a dip of 60°∠80°, while the deformed rocks on both sides have a dip ranging from 53° to 73° ∠33° to 76°, as shown in Fig 3.

## 3. Centrifugal modelling test design

The model tests were conducted using the TLJ-500 geotechnical centrifuge at Chengdu University of Technology, China. This centrifuge has a capacity of 500 g-tons and an effective radius of 4.5 m.

The dimensions of the model box were 100 cm in length, 60 cm in width, and 100 cm in height. The slope model dimensions were as follows: an inclination angle of 70°, a slope base length of 60°, and overall dimensions of 78 cm (bottom length) × 55 cm (width) × 58 cm (height), as illustrated in Fig 4(A). The final stacked configuration is shown in Fig 8I-a.

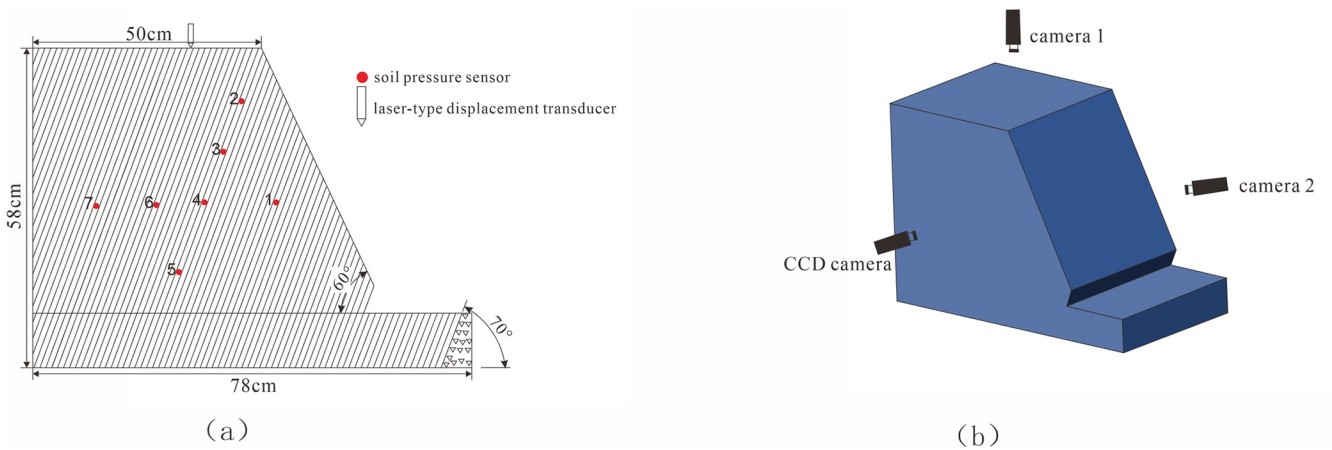

**Fig 4. Centrifugal model design.** (a) the slope model diagram, (b) Camera layout map.

**Table 1. Similarity relation of main physical quantities.**

| Quantity | Symbols | Similarity ratio Symbol | Similar constant (prototype/model) |
|---|---|---|---|
| Density | $\rho$ | $C_\rho$ | 1/1 |
| Displacement | $u$ | $C_u$ | 1/100 |
| Acceleration | $a$ | $C_a$ | 100/1 |
| Length | $L$ | $C_l$ | 1/100 |
| Stress | $\sigma$ | $C_\sigma$ | 1/1 |
| Strain | $\varepsilon$ | $C_\varepsilon$ | 1/1 |
| Poisson's ratio | $\mu$ | $C_\mu$ | 1/1 |
| cohesion | $c$ | $C_c$ | 1/1 |
| Friction angle | $\varphi$ | $C_\varphi$ | 1/1 |
| Elastic modulus | E | $C_E$ | 1/1 |

The arrangement of the soil pressure sensors, miniature cameras, and CCD industrial high-speed cameras is illustrated in Fig 4(B). Observation points were placed equidistantly on the surface to monitor and document the deformation and damage processes. Observation points were placed at equal intervals on the surface to monitor and document the deformation and damage processes, as detailed in Fig 14.

Based on the centrifuge load capacity, model box size, test objectives, and other factors, a geometric similarity ratio C1 = 1/100 (model/prototype) was selected, and the test centrifugal acceleration was 100 g. In accordance with the objectives and utilizing similarity theory, several physical quantities were selected as the primary similarity parameters, which are presented in Table 1.

The model slopes are made from similar materials, where the prototype material is a serpentine green clay millimetre, and the similar material is made from a mixture of cement, quartz sand, gypsum, and an aqueous solution of borax, and their mixed solution was used as the binding material between rock layers. After doing several multiple ratio tests, the ratio had been determined: quartz sand (28): gypsum (17): 1% borax aqueous solution (12): cement (1). The strength of the rock mass and interlayer bonding was assessed using uniaxial compressive tests and direct shear tests, with the physical and mechanical parameters of the similar materials detailed in Table 2.

The experiments were conducted using a centrifugal loading program. Initially, the centrifugal acceleration was gradually increased to 40 g and stabilized for 5 minutes. Subsequently, it was increased by 20 g increments, with each increment stabilized for 5 minutes. This process continued until reaching the target maximum acceleration of 100 g, which was then maintained for 5 minutes before being gradually decreased and stopped.

The test model was made of two types of prefabricated blocks, placed by the stacking way. The size of block A: 70 cm(length)×10 cm(width)×1 cm(thickness). The size of block B: 70 cm (length)×5 cm(width)×1 cm(thickness). Similar materials were used between layers and staggered the test blocks in the width direction to reduce the impact of main fractures and enhance communication.

**Table 2. Physical and mechanical parameters of similar materials.**

| Parameters | Density /(g.cm⁻³) | Elastic Modulus /MPa | Compressive strength /MPa | Tensile strength /MPa | Cohesion /KPa | Friction angle /(°) |
|---|---|---|---|---|---|---|
| Centrifugal model | 1.8 | 2800 | 14.8 | 1.84 | 34.7 | 42.2 |

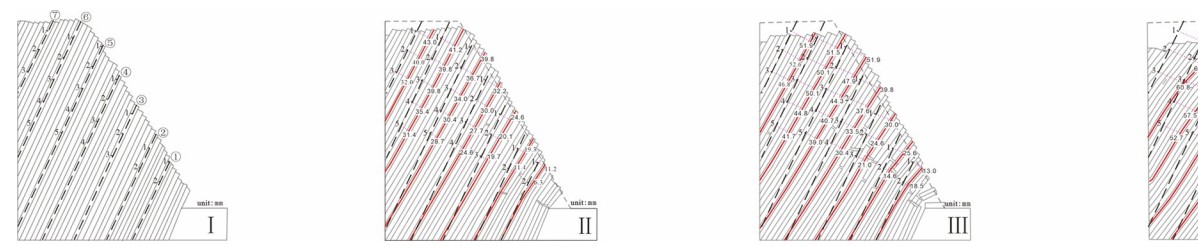

**Fig 5. Relative cumulative displacement changes at different positions.**

## 4. Analysis of the phenomenon

### 4.1 Evolution process of the toppling failure

We generated velocity contours and cumulative strike-slip displacement by comparing the image data, the centrifuge model test results, and the aging deformation between interior micro changes and surface macroscopic changes. Fig 5 illustrates the cumulative go-slip displacements in different regions. Fig 6 depicts the displacement change of each point of No. 5 rock plate. Fig 7 shows the rate of change of velocity at point 1, located on the most prominent free face.

By applying centrifugal force to the model, the complete "toppling-bending" of the slope was observed. The toppling deformation was classified into four distinct stages based on the rock sheet deformation and the expansion of the rupture: initial creep, foot toppling-bending, progressive deformation, and fracture zone penetration.

Tables 3 to 6 clearly describe the criteria for stage division and the deformation characteristics of each stage slope through the macroscopic deformation of the surface layer, the velocity contour and the cumulative displacement of each plate.

Analysis of the displacement changes on No. 5 rock plate (Fig 6) and the rate change curve at Point #1 (Fig 7), we can identify two periods of accelerated deformation and one period of stable deformation for the slope.

First Accelerated Deformation: This occurs after the slope stabilized itself. At this time, the model begins to top towards the free face. The surface macroscopic deformation of the slope is characterized by a break in the material at the foot of the slop, and the growth of fractures adjacent, creating plenty of room for the upper rocks. after the internal fracture zone penetration. During this period, the upper rocks experienced significant deformation by toppling, and the surface macroscopic deformation was predominantly characterized by the upward propagation of cracks, primarily focused on the sharp junction between the slope crest and front. A multi-level fracture zone was visible on the surface of the compacted rock slabs at the slope front, accompanied by interlayer tension cracks. The cumulative displacement increased sharply, the observation point rate of change varied significantly, and the macroscopic deformation during this accelerated stage was substantial.

Stable Deformation Process: This phase happens between the two periods of accelerated deformation. The internal rock slab bending, a tiny deformation, generated the fissures. The total displacement of rock slabs and the rates of observation stations have not been significantly affected by this.

### 4.2 Relationship between surface and internal deformation features

**4.2.1 initial creep I (Acceleration: 0-2g).** Deep layers: No significant deformation was observed; the rock slabs exhibited only slight elastic deformation.

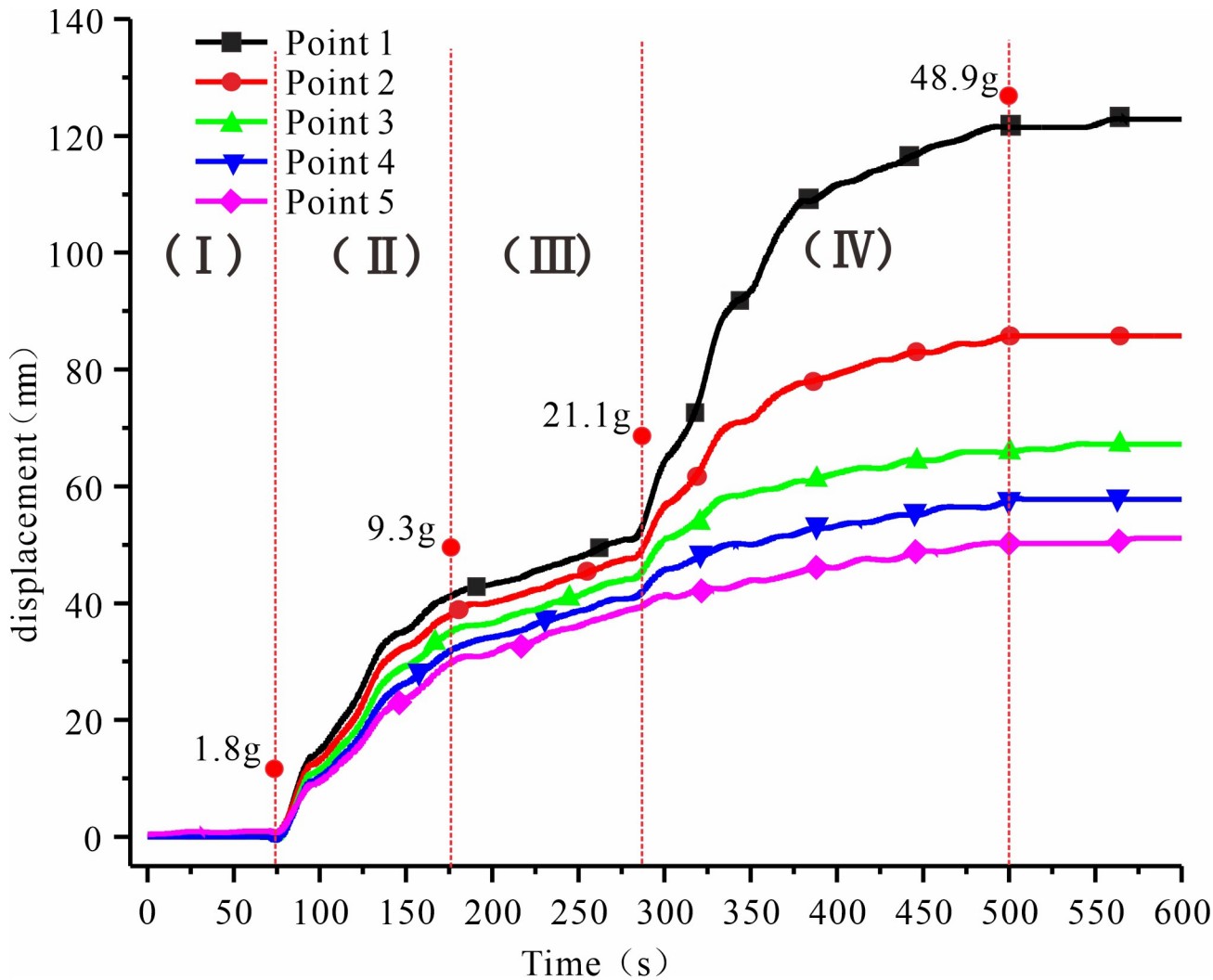

**Fig 6. Displacement change of each point on rock plate No. 5.**

Surface: Minor distortion was noted at the bottom of the slope. The local facula in the picture showed no discernible pattern, and no other components showed deformation.

**4.2.2 Foot dumping-bending II (Acceleration: 2-10g).**  Deep layers: The deepest portion of the slope was the first to fail, though it did not fully topple. At this stage, the inclination angle reached 54˚. Under tensile stress, wedge-shaped cracks formed between the rock slabs at the center of the slope front, with a maximum opening of approximately 1˚. The preceding creep had caused the slope to settle by about 5 cm at this location, and the top of the slope displayed interlayer misalignment.

Surface layer: Tension cracks appeared in the middle and lower parts of the slope, with the lower cracks becoming more pronounced and connecting to fractures at the slope base. Under continuous compression, the rock slab at the rear of the slope crest experienced interlayer misalignment and multiple crack formation. In contrast, the integrity of the front part of the slope top remained relatively good, without obvious cracks. A large deformation area, occupying approximately 1/5 of the slope's front, was focused in this region and moved at a speed of roughly 2–3 mm/s.

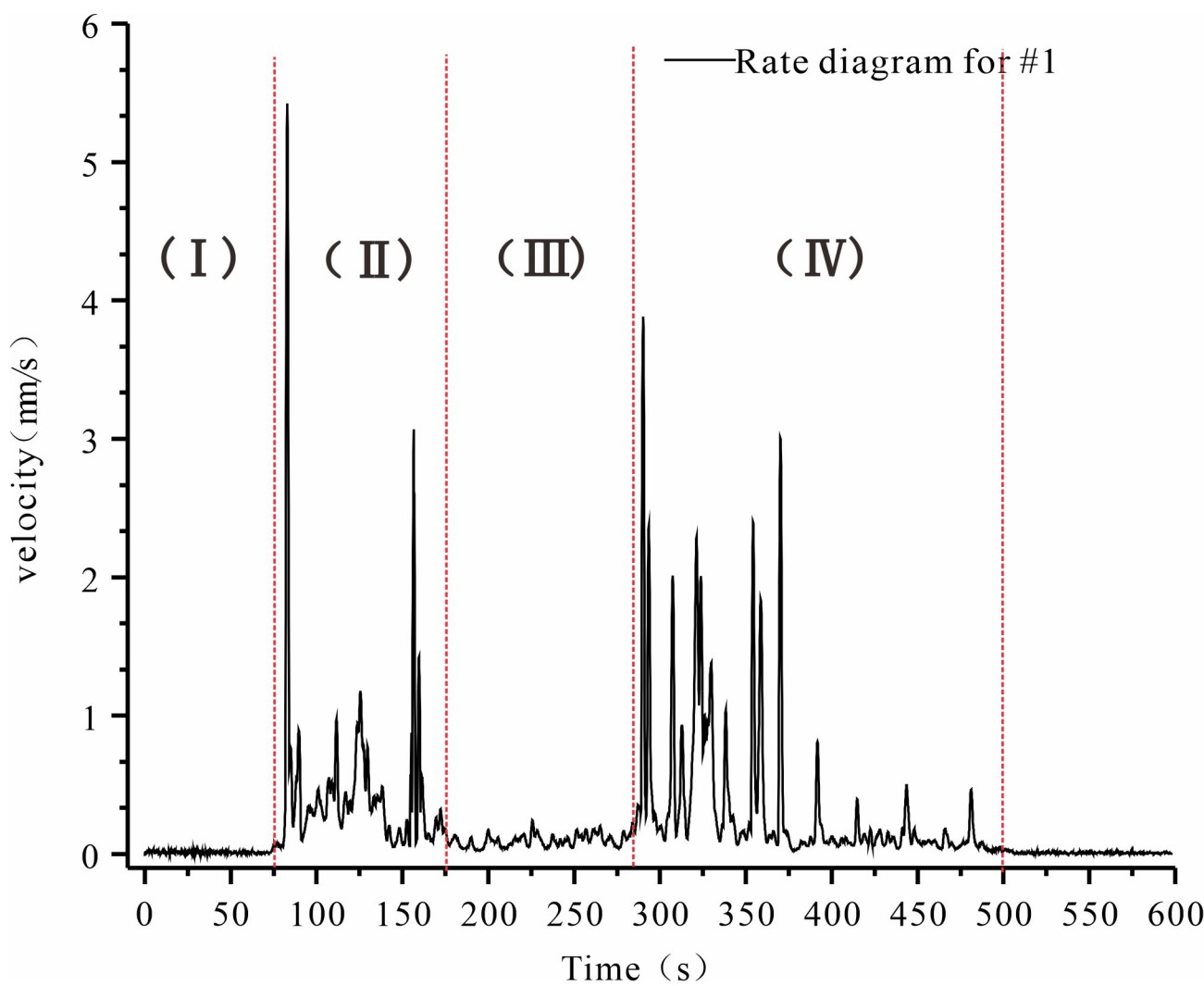

**Fig 7. Rate change curve for Point #1.**

**4.2.3 Progressive deformation III (Acceleration: 10-25g).** Deep layers: The rock slab at the base tilted and fractured towards the open side, becoming completely flat on the platform at the slope toe. The rupture surface formed by bending fracture was continuous, developing into a stepped fracture zone extending upward from the slope toe at a certain angle. A

**Table 3. The deformation characteristics of slopes in initial creep.**

| | |
|---|---|
| Signs of deformation | The maximum principal stress of the slope is parallel to the prograde near the slope toe, while the minimum principal stress is nearly orthogonal to the prograde. Under hyper gravity conditions, no significant deformation occurs in any part of the slope, and the rock slab has elastic deformation, as shown in Fig 8II-a. |
| The rate cloud diagram | In Stage I: there are no obvious signs of deformation in the deep and surface layers, though small cracks develop at the slope toe. |
| The accumulated displacement of each plate | Stage I: The slope is self-stabilizing, with no obvious signs of deformation or no displacement data, representing the initial state. |

**Table 4. The deformation characteristics of slopes in foot toppling-bending.**

| | |
|---|---|
| Signs of deformation | Under the action of a centrifugal force field, the rock within the slope was compressed as a whole. While the minor principal stress drops to zero or even changes to tensile stress, the maximum principal stress hits the highest point near the bottom of the slope. Due to stress redistribution, the internal layers inside the slope shift leftward, and the layers in front of the slope less tilt. During compaction, the rock at the back edge of the model settled and fractured due to uneven forces, which settled about 2cm and extruded about 4cm toward the front of the slope (Fig 9II-a). |
| The rate cloud diagram | Stage II: The large deformation zone is concentrated at the slope toe, and the tension between the layers is obviously due to the fracture of the rock slab at the slope toe, foot toppling-bending. Light spots appear on the surface of the rate cloud diagram. |
| The accumulated displacement of each plate | Stage II: the slope sank and topple forward. The No. 1 plate near the foot of the hill has 11.2 mm of accumulative deformation due to the toe being in the air, and there are visible tension cracks at the surface. |

secondary fracture zone then formed approximately 15 cm from the slope toe. A minor shear misalignment in front of the cracked rock slab.

Surface layers: The cover layer beneath the mid-slope section detached and accumulated at the slope toe. Due to the centrifugal force, the slabs fit more closely together, and tensile cracks formed when the tension between the rock slabs reached about 1˚. The tension crack near the top of the slope propagated further due to the ongoing creep of the model. The slope crest and mid-front area, where the velocity contours indicated major deformation, exhibited a deformation rate of approximately 2–3 mm/s.

**4.2.4 Fracture zone penetration IV (Acceleration: 25-50g).** Deep layers: The slabs at the slope toe were fractured sequentially, with multi-level fracture zones running through; The first fracture zone cut through the top of the slope, creating a fracture with a depth of approximately 200 mm and an inclination of about 38˚ in the upper rock layer. The second fracture zone running through the front of the slope, the inclination of the upper slab decreased further about 34˚. After slope destabilization the adhesion between the slab in front of the slope was deeper and created a tension crack with a 1˚ crack at the slope's top.

**Table 5. The deformation characteristics of slopes in progressive deformation.**

| | |
|---|---|
| Signs of deformation | The rock slab initially fractured due to the maximum principal stress difference near the bottom of the slope. After the fracture the rock slab slid forward by around 0–1mm. At the bottom of the slope, the rock above the layer loses essential support, and the deformation space expands. The toppling deformation is followed by a steady decrease in toppling degree as it approaches the slope's summit under supergravity. The first fracture plane originated at the slope toe, expanding upward at a certain angle and forming a second fracture zone about 15 cm from the foot. This second zone extended upward to 20 cm without further expansion. The free surface has been pushed forward by about 5.5 cm and the slope model has settled down about 6 cm at this point. (Fig 10II-a) |
| The rate cloud diagram | Stage III: The large deformation zone shifts upward to the middle of the slope. Due to the overturning and compacting of the rock slab at the slope toe, the tension crack between the layers in the middle of the slope developed. Large-scale spalling and accumulation of the cover layer in front of the slope; |
| The accumulated displacement of each plate | Stage III: the slope continues to sank slowly, and the deformation in each slab increases gradually. The tension cracks that form in the center of the sloping front cause the No. 1 rock slab to shatter, increasing its deformation to 13 mm. The No. 3 rock slab's total displacement increased to 30 mm. The No. 5 rock slab increases to 51.9 mm, which is the maximum accumulative displacement in this stage, whose surface cracks are developed. |

   

**Table 6. The deformation characteristics of slopes in fracture zone penetration.**

| Signs of deformation | The interlayer tension and shearing effects along the cut layers intensify as the rupture surface forms. The sheet in front of the model remains bent, while the top of the first fracture plane and the front of the slope of the second fracture zone are both visible. The distance from the top of the first fracture to the bottom of the first fracture plane is 200 mm. After the inclination above the second fracture zone is reduced, the slab is extruded from the lower portion of the slope, as shown in Fig 11II-a. |
|---|---|
| The rate cloud diagram | Stage IV: The large deformation zone is moves up to the steep junction, where the upper rock slab overturns as a whole due to the breakthrough of the fracture zone. The deformation of the steep junction is the largest due to the influence of the free face. Large surface deformations are concentrated at the steep junction between the top and front of the slope, with smaller cracks developing on the surface. |
| The accumulated displacement of each plate | Stage IV: The fracture zone penetrates s, causing significant damage thought toppling. The accumulative deformation of the No. 1 rock slab is 21 mm, roughly twice as much as in the previous period. The No. 3&4 rock slab's total displacement slowly increases while it got loaded. The second fracture zone shares the upper inclination angle of plates 5 and 6 and causes the highest deformation, extending approximately 121 mm and 106 mm. This area becomes the largest deformation zone in this stage. |

Surface layers: The middle and lower rock slabs were compressed under pressure while the rock slabs in front of the slope were dumped and piled up. Shear damage affected some of the slabs, causing them to crack. The tiny cracks developed at the toe of the top of the slope.

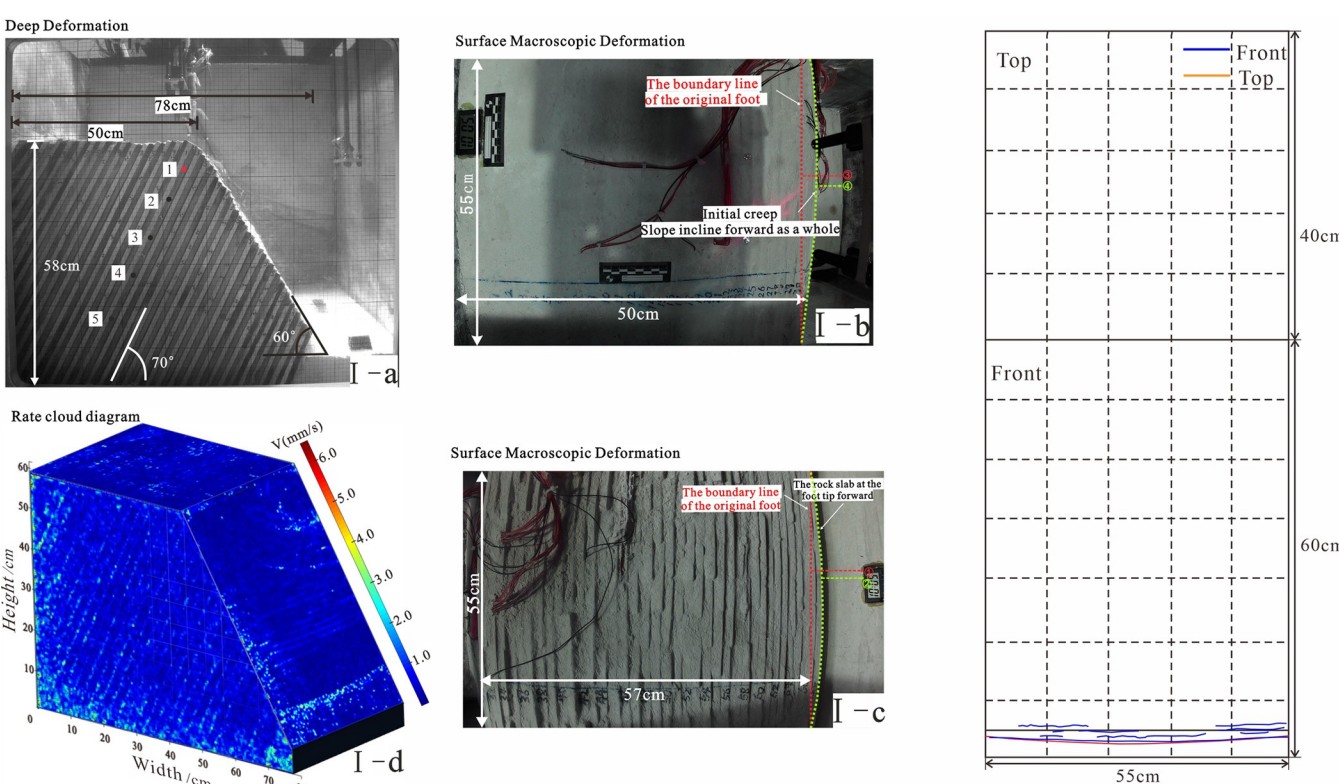

**Fig 8. The relationship between deep deformation and surface macroscopic deformation in stage 1.** Note: 1. The b-plot of each stage is the top of the slope, the c-plot is the front of the slope, the d-plot is the velocity cloud, and surface fracture development. 2. The numbers in the rate cloud diagram refer to the average rate of total displacement change within a given stage.

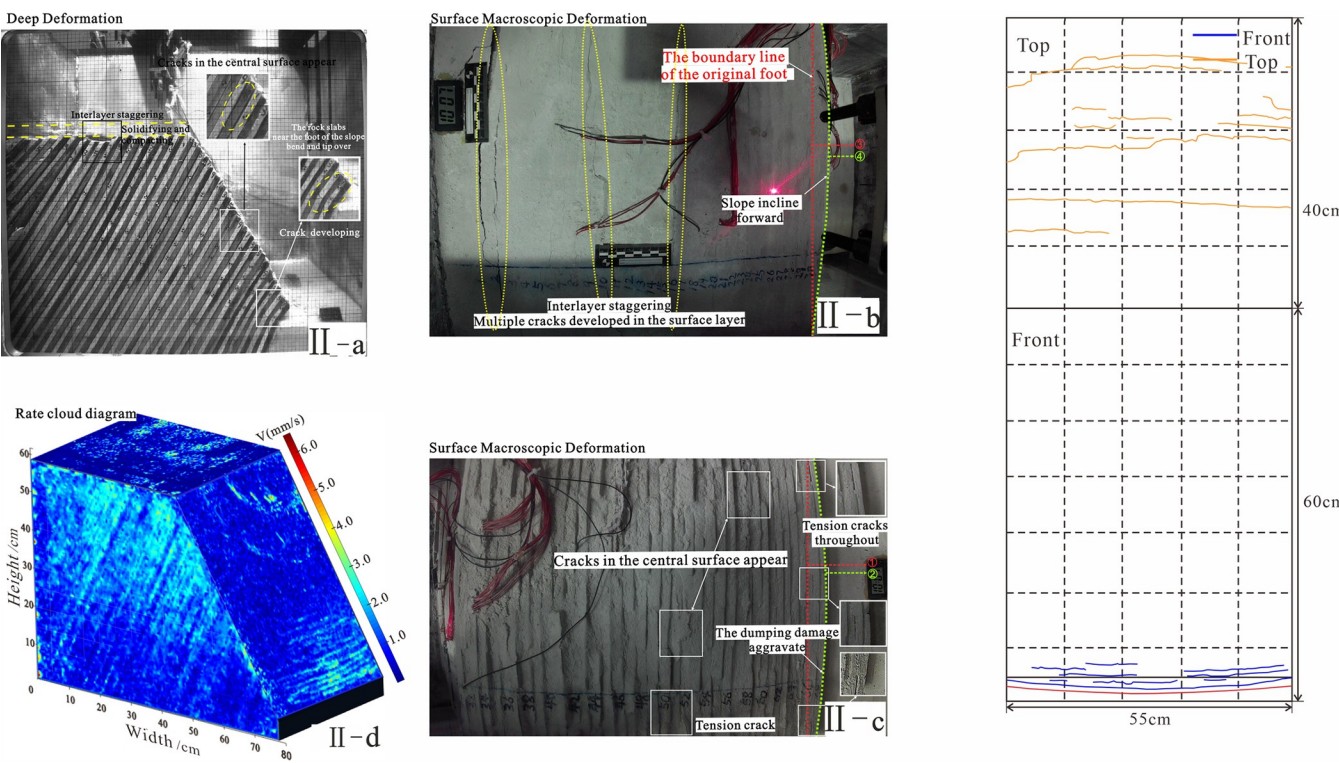

**Fig 9. The relationship between deep deformation and surface macroscopic deformation in stage 2.**

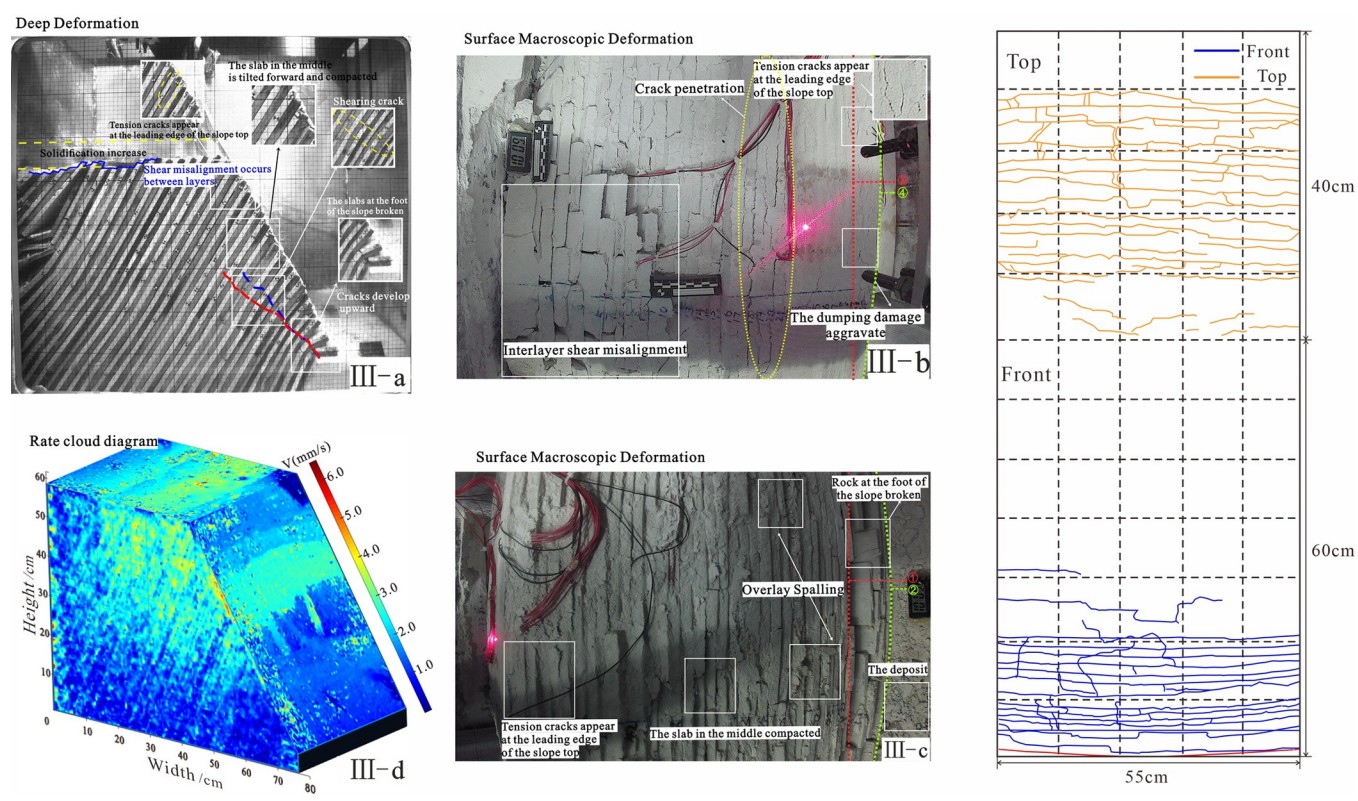

**Fig 10. The relationship between deep deformation and surface macroscopic deformation in stage 3.**

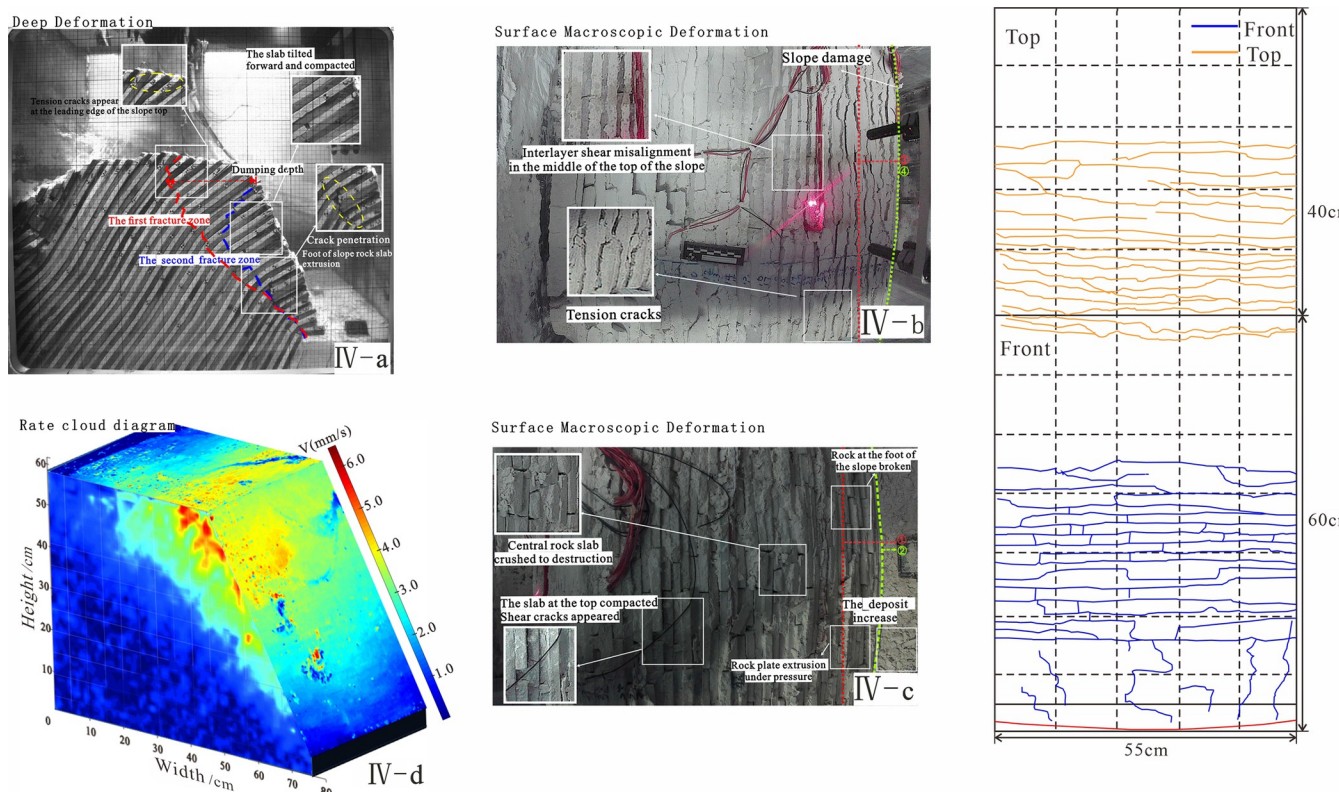

**Fig 11. The relationship between deep deformation and surface macroscopic deformation in stage 4.**

According to the centrifugal model destabilization process, the macroscopic deformation characteristics of the surface layer on the anti-dip stratiform rocky slopes can be summarized as follows. The major deformation areas on the slope surface correspond to locations where key cracks develop. During the slope destabilization process, these deformations progress through three stages: from the slope toe to the middle and upper parts of the slope front, and finally to the top where the slope transitions from steep to gentle. This phenomenon is clearly correlated with the development process of the internal rupture surface. Fig 12 shows the development pattern of the internal and surface cracks of the slope.

## 5. Theoretical analysis

### 5.1 Analysis of fracture zone location

The fracture zone of the slope starts from the foot to the top in the form of steps. The fracture zone of the slope started from the foot to the top in the form of steps.

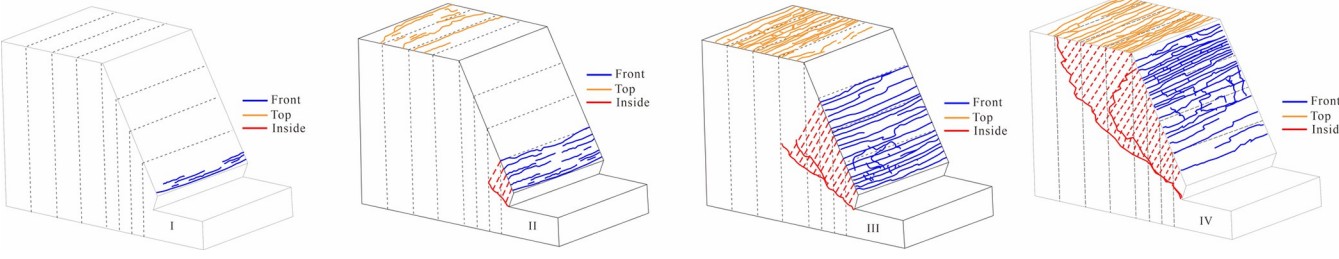

**Fig 12. Crack development at different positions.**

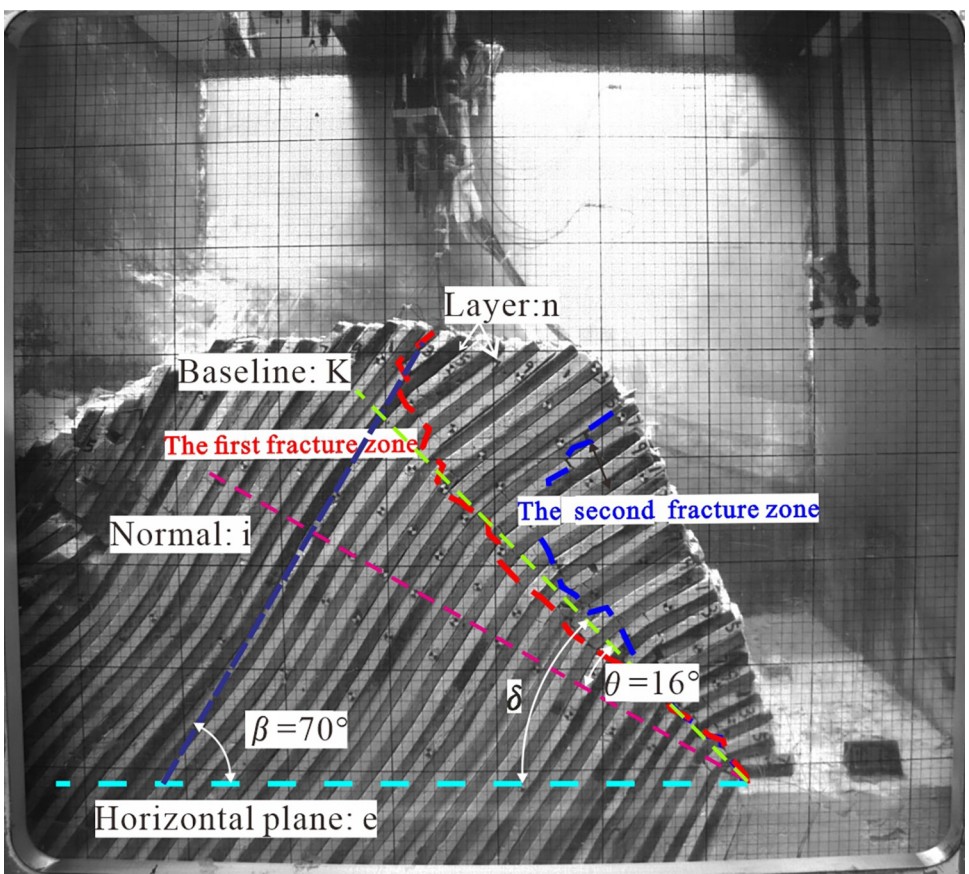

**Fig 13. Analysis diagram of broken belt position.**

From the profile view, the fracture zone approximates a straight line, represented by a straight line *K* across the foot of the slope, referred to as the baseline of the dumping fracture (Fig 13).

Above line *K* is the toppling area, and the below area is not deformed. The angle between the baseline *K* and the level normal *i* is denoted by $\theta$, in this test $\theta = 16°$. This conclusion is consistent with Adhikary's conclusion that the angle between the reference plane and the level normal lies from 12° to 20°. When the inclination of the rock layer is $\beta$, the angle between the baseline *K* and the horizontal plane e is known as (1) (Fig 14):

$$\delta = \theta + 90° - \beta \qquad (1)$$

Prerequisites are necessary for the anti-dip stratiform rocky slope to discharge damage in its natural form; only the rock layer above the baseline can become unstable. It can be inferred that: for anti-dip stratiform rocky slopes, the magnitude of $\delta$ can be calculated by Eq (1) when the dip angle of the rock formation is $\beta$ If the slope angle $\alpha > \delta$, the slope will be deformed by toppling above the toppling fracture zone. In this study, the slope angle is $\alpha = 60° > \delta = 36°$, and toppling occurs, achieving the expected test effect. The stability of the slope will be determined by the fracture zone's degree of development when determining the slope's stage.

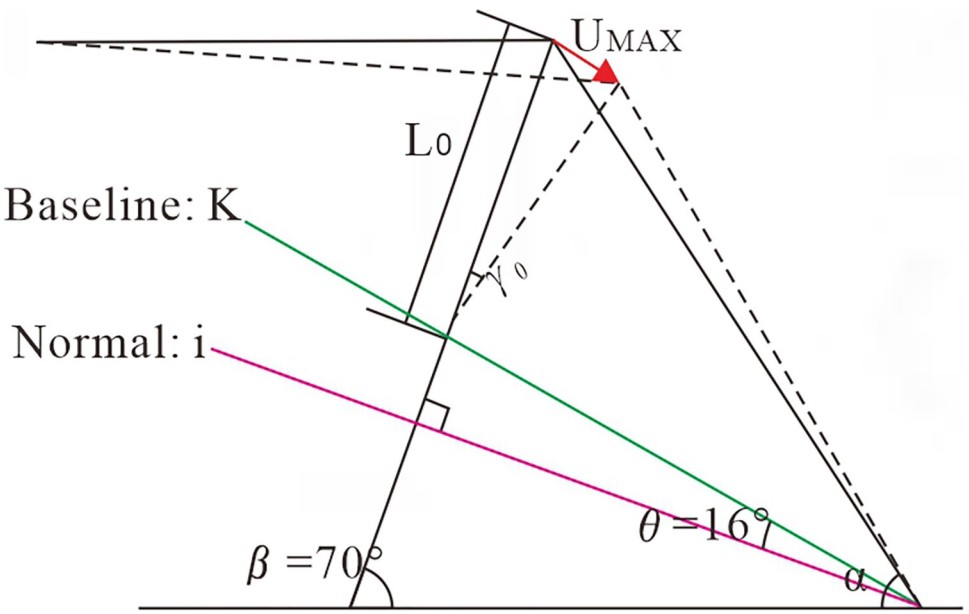

**Fig 14. Cumulative displacement calculation diagram of the steep and gentle junction.**

## 5.2 Cumulative displacement thresholds on the judgment of toppling

The degree of fracture zone development, while useful for identifying the stage of slope failure, is less practical for determining slope stability. The trend of the cumulative displacement curve of No. 5 rock is very consistent with the destruction process of the slope, according to the analysis of experimental phenomena and data. Using the variation characteristics of the cumulative displacement curve in conjunction with the rate variation at the observation point of the optimal critical conditions and referring to the actual deformation of the surface layer, the overall stability of the slope can be evaluated. Combine with the flexural equation, to build the calculation method of the slope toppling deformation threshold $U_{MAX}$. According to the maximum deformation ($U_{MAX}$) that the slope can withstand in the process of deformation and destruction, and combined with the actual displacement of the slope at this time can reasonably determine the deformation stage of the slope at this time, and have an accurate understanding of the stability of the slope [28].

From Fig 14, a special geometric relationship exists between the free cantilever $L_0$ at the steep and gentle junction and the slope rupture surface. According to certain geometric conversions, the value of $L_0$ can be expressed by the following equation:

$$L_0 = \frac{H sin(\alpha + \beta - \theta - 90°)}{sin\alpha \; sin(90° - \theta)} \tag{2}$$

$H$ is the height of the slope; $\alpha$ is the slope angle; $\beta$ is the dip angle of the rock layer; $\theta$ is the angle between the slope fracture baseline and the level normal, which is 16˚.

Analyzing the rocky beams at the steep junction separately and treating the anti-dip stratiform rocky slope as a collection of cantilever beams put one above the other. Since the gravitational load on the slope increases progressively from the top downward, it is reasonable to assume that this load acts on each cantilevered rock slab as a triangular, linearly distributed load, as illustrated in Fig 15.

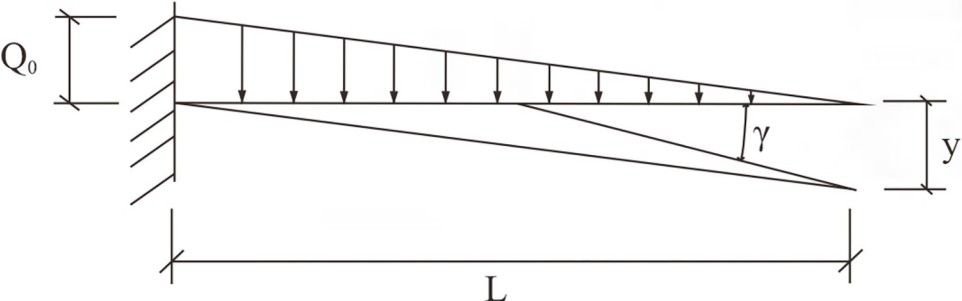

**Fig 15. Mechanical analysis diagram of a simple cantilever beam.**

According to the basic principles of structural mechanics, the deflection and rotation angle of a cantilever beam under a triangle load with linear distribution is depicted in the following Eqs (3) and (4). $\gamma$ and $y$ are the angle of rotation (rad) and deflection (m) of the cantilever beam under the triangular linearly distributed load respectively, $Q_0$ is the maximum value of the triangular linearly distributed load acting on the cantilever beam, L is the length of the cantilever section of the beam; $E$ and $I$ are the modulus of elasticity and stiffness of the cantilever beam respectively. The formula's positive and negative signs, which stand for clockwise and counterclockwise rotation, respectively, indicate the direction of the beam's rotation.

$$\gamma = -(Q_0\,L^3)/24EI \tag{3}$$

$$y = -(Q_0\,L^3)/30EI \tag{4}$$

Combining Eqs 3 and 4, under the triangular linearly distributed load has a relationship with the deflection and angle of rotation.

$$y = 4\gamma L/5 \tag{5}$$

The maximum deformation of the toe part of the slope can be found from the length $L_0$ of the cantilever section of the rock slab and the maximum dip angle $\gamma_{cr}$ of the rock formation. The toppling displacement $U_{MAX}$ is expressed as:

$$U_{MAX} = (4\gamma_{cr}\,L_0)/5 \tag{6}$$

According to the above centrifugal test, combined with Huang Runqiu's deformation system [29], the toppling angle $\gamma_{cr}$, slope toppling until destabilization, is taken as 45°. Combine Eqs 2 and 6, this ultimate displacement $U_{MAX}$ can be expressed as:

$$U_{MAX} = 4\gamma_{cr}\frac{H sin(\alpha + \beta - 90° - \theta)}{5\,sin\alpha\,\sin(90° - \theta)} \tag{7}$$

The $U_{MAX}$ derived from Eq 7 represents the limiting value of slope deformation from the beginning to the critical damage. If the cumulative displacement at the free face does not surpass the maximum deformation $U_{MAX}$, the slope will not be destroyed. The indicators of this test are substituted into Eq 7, we get $U_{MAX}$ = 17.4 cm, this number is the cumulative displacement of the rock slab close to the surface at the steep and gentle junction. For the previously analyzed point 1, it is also necessary to subtract 5 cm, $U_{1MAX}$ = 12.4 cm. The inference regarding $U_{MAX}$ is supported by the fact that the theoretical findings are comparable to the maximum value of the cumulative displacement curve at point 1 (Fig 6) in the experiment.

### 5.3 Coefficient of toppling bending state

Goodman and Bray proposed that the overturning deformation of layered slope rock mass can be summarized as three types [9], in order to describe the degree of toppling deformation of each block when the landslide occurs, the toppling bending state coefficient $Df$ is defined.

$$D_f = V_T / V_B \tag{8}$$

Where $D_f$ is the coefficient of the bending state of the slope, $V_T$ is the average velocity of the slab at point T during the toppling process, $V_B$ is the average velocity of the slab during the deformation process at point B, point T is the point at the top of the slope, and point B is the point at the bottom of the slope.

In the experimental model, 44 monitoring points were established with the longitudinal profile as the datum, which were divided into five areas from the surface of the slope to the deep part and from the slope foot to the top of the slope (Fig 16). The motion characteristics of the corresponding points were monitored, and the tilt bending state coefficient $D_f$ was calculated (Fig 17).

When $D_f \approx 0.5$, the point is located near the fracture zone, which is sheared and broken, which is the development area of the fracture zone. When $D_f > 0.5$, the point is located in the upper part of the fracture zone, and the rock slab is bent and broken and overturned forward,

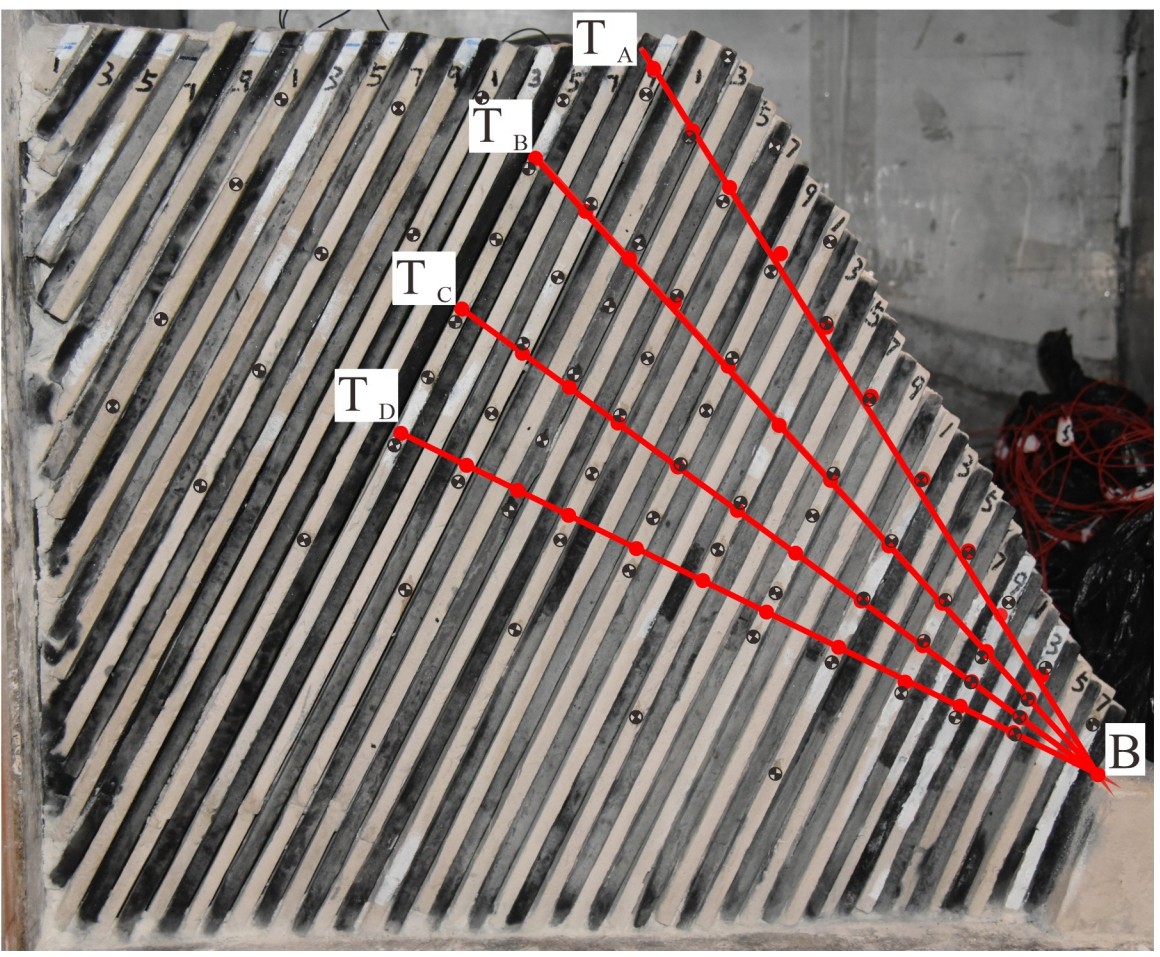

**Fig 16. Identification point selection.**

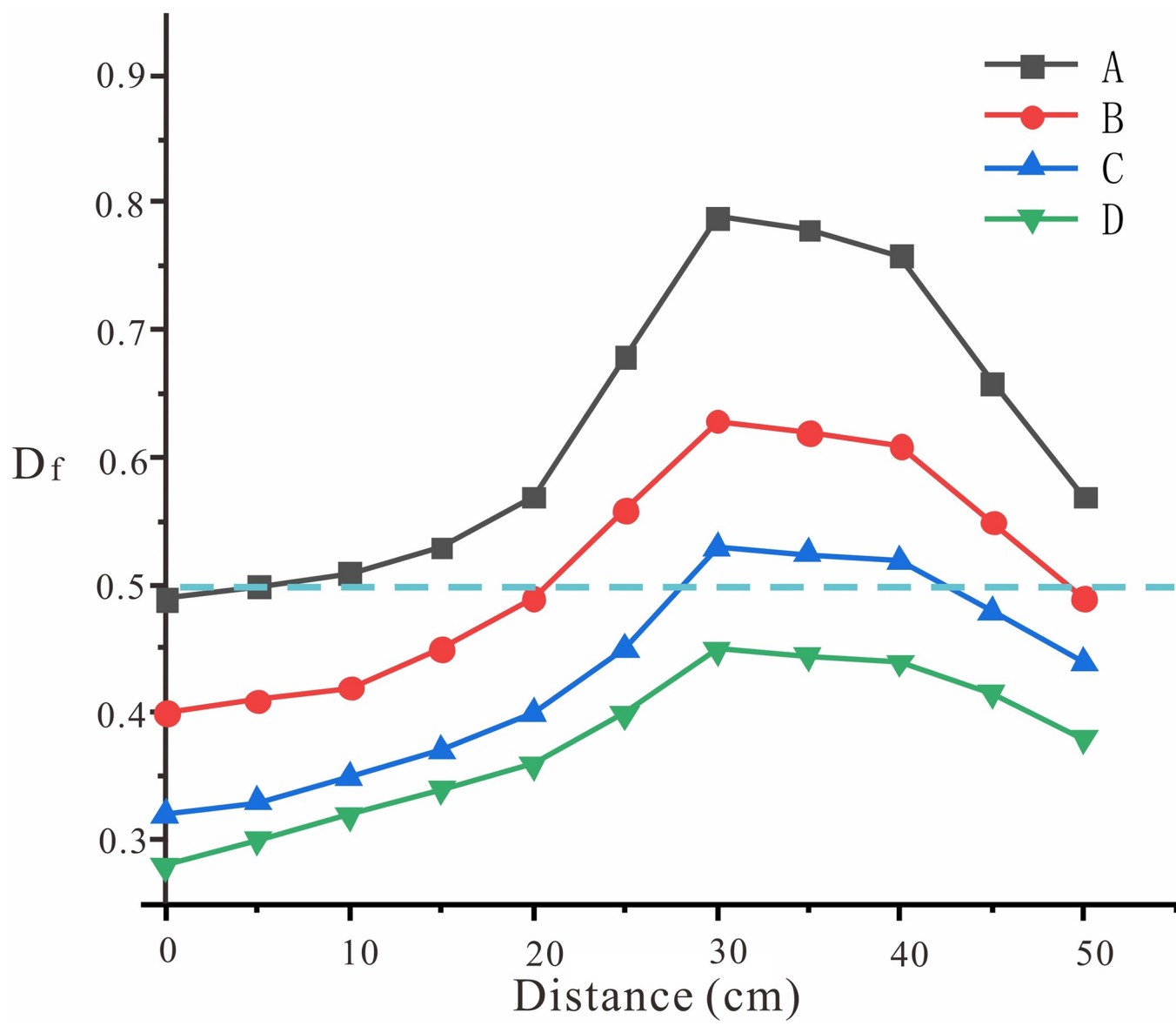

**Fig 17. Coefficient of toppling bending state.** Df.

which is a large deformation area, and when the $D_f < 0.5$, the point is in the lower part of the fracture zone, and the rock slab is only slightly bent and compacted, and the whole is relatively stable, which is a stable area. In Fig 17, the horizontal axis "Distance" refers to the distance between the selected point T and the toe of the slope point B.

The toe area is a significant shear slip zone where stress is concentrated within the rock strata Shearing and overturning occur following the increased exposure of the foot of the slope to free conditions. The largest bending zone is located in the middle of the slope, within 40–60 cm from the foot of the slope, and the significant overturning area is located above the fracture zone, as shown in Fig 16 above Zone B-$T_B$. This part of the rock mass mainly undergoes toppling deformation, which is characterized by multi-layer bending and overturning deformation, rock mass fragmentation, and shear sliding due to the loss of support in the lower part of the fracture zone.

**Fig 18. Toppling deformation disaster mode diagram.**

## 6. The relationship between the surface and internal deformation features of anti-dip stratiform rocky slopes

The "toppling and bending" deformation was mainly developed in inwardly inclined laminated rocky slopes. The cantilever beam was initially bent towards the free face by the self-weight of the inward-dipping rock mass at the slope toe. This bending gradually progresses inward along slope. Cracks also developed on the surface of the body that had been toppled, generating a counter-slope step parallel to the alignment and accompanying the bent girders' misalignment with one another (Fig 18). The following is a specific analysis of the coupled relationship between these factors.

1. The period of initial creep: The slope initially self-stabilized with minor folding and rising of the overburden occurring near the slope toe. There were no noticeable signs of deformation in both the surface and deep parts. Localized small cracks developing at the foot of the slope, no transverse penetration was observed.

2. The period of foot toppling-bending: The period of foot toppling-bending: The area of strong toppling deformation was concentrated at the slope toe (Fig 19–II) because the maximum deviator stress ($\sigma_1 - \sigma_3$) was largest near the toe. This resulted in toppling-bending of the rock slabs at the slope toe. The tension cracks between layers on the surface were developed, transversely penetrating, with a certain degree of tension, and the overburden layer was locally slipping and piling up. The center and rear rocks became misaligned, creating a counter-slope step t parallel to the bedding.

3. The period of progressive deformation: The area of strong deformation developed toward the middle of the slope (Fig 19-III), the cracks at the slope toe were compressed and compacted by the upper rock slab. Key tension cracks in the mid-slope section front were developed and transversely penetrated, while interlayer cracks at the toe of the slope top were developed. The overburden layer in front of the slope experienced significant spalling and accumulation. The tension cracks between the layers of the ground surface grow as the rock slab at the top of the slope loses the effective support of the central rock body and dips

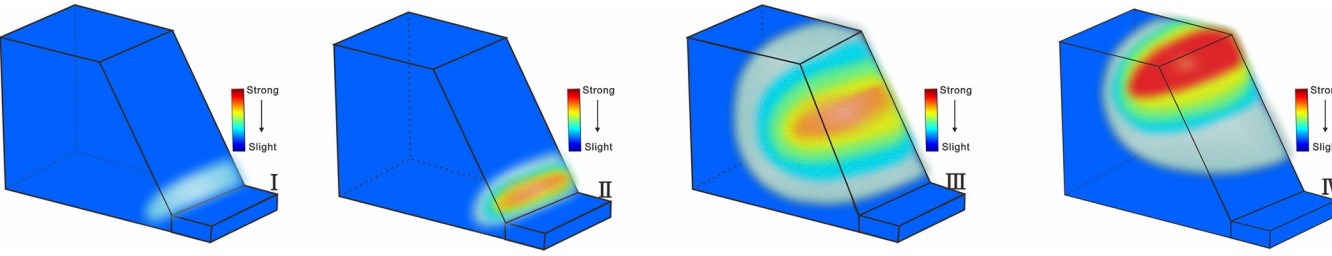

**Fig 19. Distribution of the severity of toppling deformation.**

forward. 4) The steep intersection where the slope's front and the top meet was where the area of significant deformation was located (Fig 19-IV). The entire rock body in the upper part of the fracture zone dips forward and compacts. The key fracture was located at the location of the fracture zone penetration in the middle of the slope top, and smaller fractures were densely developed at the surface. If a secondary fracture zone existed, its penetration to the surface would also be accompanied by the development of critical fractures. As a result, the location of the development of the superficial key fracture could be used as a demarcation criterion to classify the stage of dip deformation of the anti-dip stratiform rocky slope in addition to the development process of the rupture surface.

The rate fluctuation pattern at the steep-gentle junction in Fig 7 was compatible with the aforementioned macroscopic deformation properties. The first peak occurs in stage II, when the slope whole began to slump and deformation initiates under optimal proximity conditions. This stage exhibited the highest degree of rock slab overturning and the highest rate of variation in its area as a result of the ideal proximity conditions. The second peak occurred in stage IV, when the internal fracture zone was broken through, and the multi-level fracture zone shears off the rock slab at the steep junction.

In summary, the evolutionary process of the large deformation area. The stage classification of the dip-deformation pattern of anti-dip stratiform rocky slopes used the position of the development of important cracks in the surface layer as a standard. This approach not only allowed for the observation of toppling deformation but also facilitated the integration of deformation data with InSAR and other relevant techniques. Such integration aids in accurately identifying the stage of slope deformation and provides crucial support for subsequent management and preventive measures.

## 7. InSAR deformation characterization of the deformation zone

The C-band Sentinel-1A radar satellite images, provided by ESA, along with the archived data from the Wide Area Surface Deformation Detection (WASSD) satellite, offer valuable resources for this study. The WASSD satellite, with its 12-day revisit period and high temporal resolution, provides freely accessible data. For this analysis, elevation orbit images from the Sentinel-1 IW mode were utilized, which include 409 scenes of ascending orbit data and 308 views of descending orbit data. The details of the selected Sentinel-1A lift-orbit data used in this paper are summarized in Table 7.

The Stacking-InSAR technique can effectively reduce the error in Differential InSAR technique, and can better obtain the deformation range and the morphological characteristics of deformation in space. This approach leverages weighted averaging of stacked phases to obtain detailed spatial deformation characteristics. The data collected from Sentinel-1, including 409 scenes of up-orbiting data from January 2018 to October 2022 and 308 scenes of down-orbiting data from January 2018 to November 2022, were processed. The results, showcasing surface deformation during the monitoring period, are presented in Fig 20.

Table 7. Sentinel-1 up-orbiting and down-orbiting data parameters.

| SAR Sensor | Orientations | Heading angle (°) | Angle of incidence (°) |
|---|---|---|---|
| Sentinel-1A | up-orbiting (path 99) | -12.76 | 36.7 |
| Sentinel-1A | down-orbiting (path 33) | 192.72 | 39.6 |
| Sentinel-1A | down-orbiting (path 106) | 192.85 | 36.9 |

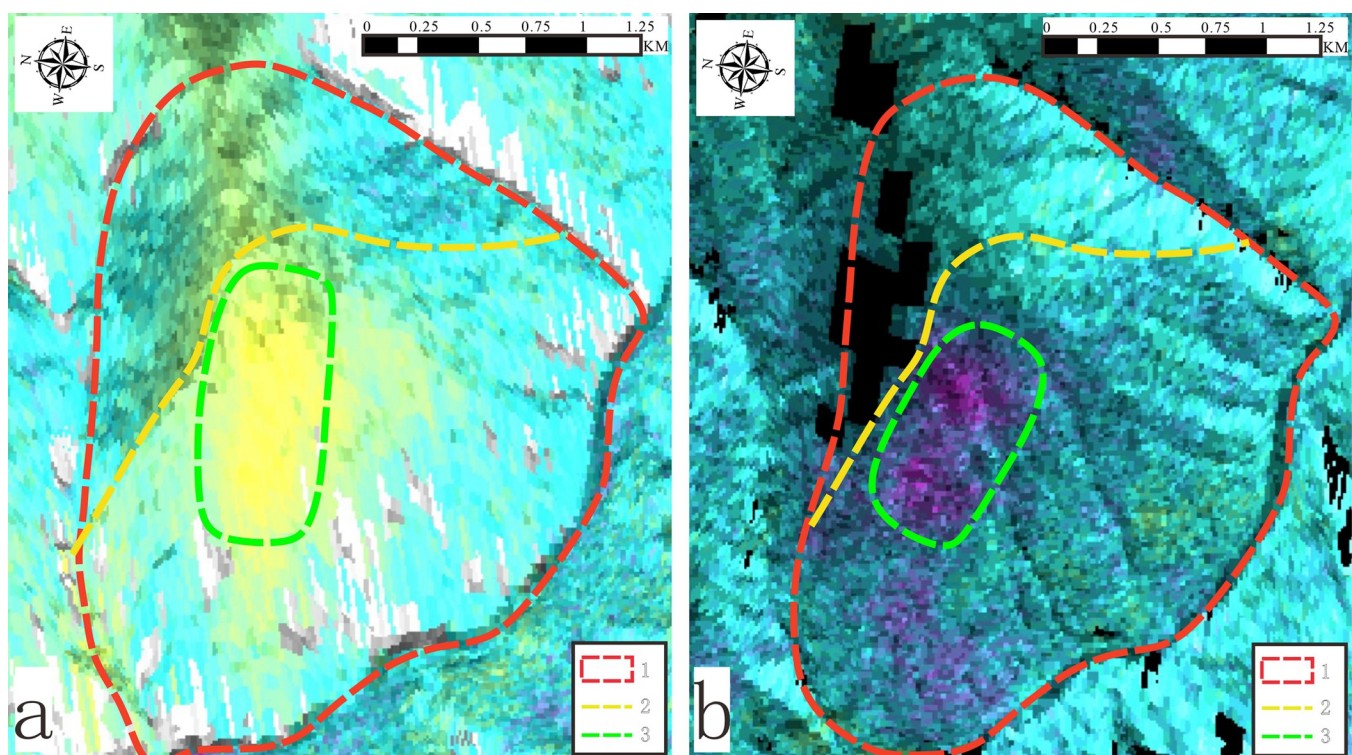

**Fig 20. Stacking-InSAR derived deformation rate maps. 1.** Deformation boundary range **2.** Steep and gentle junction **3.** main deformation zone. (a) ascending orbit, (b) descending orbit.

According to the decoding results, the majority of the deformed body and its steep intersection with the slope's top is where the majority of the deformation is focused. The field investigation a attributed this to the excavation of the slope for road construction, as well as the significant deformation of the exposed bedrock during the geological evolution process. Fig 21 is a drone image. The slope is currently stable overall, but it still includes shear slip, tension rupture, fracture-fall overlap, and other phenomena.

The Zha Yong toppling deformation is in the III stage of slope toppling deformation (progressive deformation stage) and has the potential to advance to the next stage, according to the practical field investigation and thorough analysis of the research findings.

The main surface macro features at this stage are: The rock slab in the mid and lower slope are bent and compressed, the rupture surface is produced and propagating upward, and interlayer tension cracks are developed at the steep and gentle junction, and its large deformation area focuses on this. These features collectively illustrate the ongoing deformation processes affecting the slope, emphasizing areas of critical instability and deformation.

Comparison of the field investigation and the InSAR remote sensing interpretation results verified the accuracy of the study on the deformation pattern of inclined rocky slopes and the surface layer's macro deformation characteristics, as well as the rationality of the deformation phases. This proves that the surface macro-deformation characteristics (i.e. the development of key cracks) can also be used for deformation monitoring. By combining the rate change characteristics of the points at the optimal critical conditions and the cumulative displacement changes of the rock slabs, a slope deformation model was established.

The corresponding coupling relationship between the slope deformation pattern and the surface macro-deformation characteristics was established by integrating the rate change and

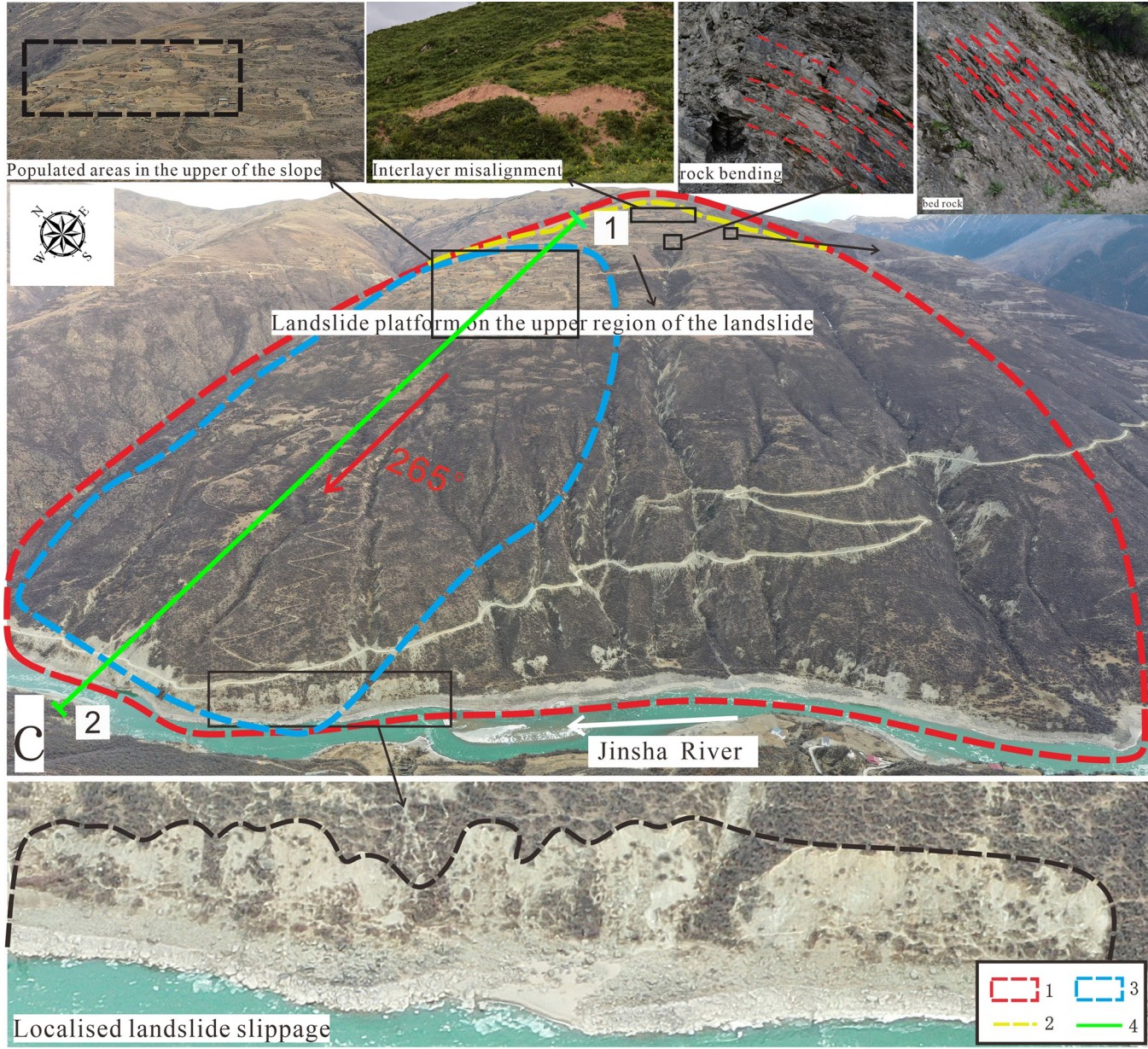

**Fig 21. Field survey of the Zha Yong toppling deformation. 1.** Deformation boundary range **2.** Steep and gentle junction **3.** main deformation zone **4.** profile line.

cumulative displacement data. By combining the InSAR remote sensing interpretation with the actual ground investigation, the stage of the slope can be deduced in reverse. This method provides valuable technical support for early identification, prevention, and control of major landslide disasters.

## 8. Conclusions

This study examines the temporal between the surface layer's macroscopic deformation properties and the toppling deformation mode of anti-dip rocky slopes. Large-scale centrifugal model testing was conducted, and the monitoring data and test images were compared and

analyzed. Using the Zhayong toppling deformation upstream of the Jinsha River as a geological prototype for comparison. The following conclusions were obtained:

1. The bending and toppling failure is mainly divided into four stages: initial creep, foot toppling-bending, progressive deformation, and fracture zone penetration. Failures typically initiate at the slope toe, progress towards the middle and upper sections, and ultimately lead to the complete failure of the slope as the stepped fracture surface develops. The large deformation area is mainly concentrated at the steep junction between the toe of the slope and the top of the slope.

2. Calculating the cumulative displacement at optimal critical points at the steep and slow junctions helps determine the critical threshold values and their evolution during slope failure. This approach provides a reliable method for dividing slope deformation stages and effectively monitoring the deformation progress.

3. In the process of slope toppling, the surface macroscopic deformation was analyzed to establish the coupling relationship between toppling deformation mode and macroscopic deformation. This allowed proposing the location of key fracture development on the surface layer as an effective tool to classify the damage phase of slope toppling. The key cracks developed in the following order: near the foot of the slope—the middle of the front of the slope—the steep junction at the top of the slope.

4. The InSAR data for the Zayong deformation were analyzed alongside ground investigation results to assess macro-deformation characteristics. By reverse extrapolating the deformation stages, it was determined that the Zayong deformed body was initially in the third stage of collapse damage and progressing towards the fourth stage. Although there is a tendency for the slope to continue developing, it remains stable for now because the fault zone has not yet fully penetrated. This process visualizes the stage of slope deformation and provides technical support for the identification, prevention and control of such slopes.

## Supporting information

**S1 Data.**
(RAR)

**S2 Data.**
(RAR)

**S3 Data.**
(RAR)

**S4 Data.**
(RAR)

**S5 Data.**
(RAR)

**S6 Data.**
(RAR)

**S7 Data.**
(RAR)

**S8 Data.**
(RAR)

**S9 Data.**
(OPJU)

## Author Contributions

**Conceptualization:** Jun Chen.

**Data curation:** Minghao Chen, Huiyan Lu.

**Funding acquisition:** Guang Zheng.

**Resources:** Qi Liu.

**Supervision:** Guang Zheng.

**Validation:** Jinning Yang, Jun Chen.

**Writing – original draft:** Mingyu Zhang.

**Writing – review & editing:** Guang Zheng, Mingyu Zhang.

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
