## [Decision Letter · Decision Letter 0]

30 Jul 2024

PONE-D-24-28956Deformation stage division based on the surface behavior characteristics and internal rupture characteristics of toppling  failure kinematic featuresPLOS ONE

Dear Dr. Zhang,

Thank you for submitting your manuscript to PLOS ONE. After careful consideration, we feel that it has merit but does not fully meet PLOS ONE’s publication criteria as it currently stands. Therefore, we invite you to submit a revised version of the manuscript that addresses the points raised during the review process.

We look forward to receiving your revised manuscript.

Kind regards,

Linwei Li

Academic Editor

PLOS ONE

Journal Requirements:

The name of the colleague or the details of the professional service that edited your manuscript.A copy of your manuscript showing your changes by either highlighting them or using track changes (uploaded as a *supporting information* file).A clean copy of the edited manuscript (uploaded as the new *manuscript* file).

"This study is funded by the National Key R&D Program of China (No. 2021YFC3000401)."

Please note that funding information should not appear in the Acknowledgments section or other areas of your manuscript. We will only publish funding information present in the Funding Statement section of the online submission form. Please remove any funding-related text from the manuscript. 

"This study is funded by the National Key R&D Program of China (No. 2021YFC3000401)."

5. We note that your Data Availability Statement is currently as follows:

"All relevant data are within the manuscript and its Supporting Information files."

7. Please ensure that you refer to Figures 1, and 3 in your text as, if accepted, production will need this reference to link the reader to the figure.

8. We note that Figures 18, 19, and 21 in your submission contain satellite images which may be copyrighted. All PLOS content is published under the Creative Commons Attribution License (CC BY 4.0), which means that the manuscript, images, and Supporting Information files will be freely available online, and any third party is permitted to access, download, copy, distribute, and use these materials in any way, even commercially, with proper attribution. For these reasons, we cannot publish previously copyrighted maps or satellite images created using proprietary data, such as Google software (Google Maps, Street View, and Earth). For more information, see our copyright guidelines: http://journals.plos.org/plosone/s/licenses-and-copyright.

1) You may seek permission from the original copyright holder of Figures 18, 19, and 21 to publish the content specifically under the CC BY 4.0 license.  

2) If you are unable to obtain permission from the original copyright holder to publish these figures under the CC BY 4.0 license or if the copyright holder’s requirements are incompatible with the CC BY 4.0 license, please either i) remove the figure or ii) supply a replacement figure that complies with the CC BY 4.0 license. Please check copyright information on all replacement figures and update the figure caption with source information. If applicable, please specify in the figure caption text when a figure is similar but not identical to the original image and is therefore for illustrative purposes only.

**Additional Editor Comments:**

The review process is now complete, and we have three reports submitted by the expert reviewers. As can be seen from these reports, two reviewers requested revision before the manuscript was accepted. However, one of them recommended the rejection of the manuscript. The author must be aware that the comments of reviewers #1 and #3 are serious. Therefore, I suggest the authors carefully revise this manuscript point by point according to all the comments. The resubmitted manuscript will be reviewed again.

In the end, I want to thank the reviewers for their great efforts on your manuscript and you for your submission.

All the best,

Linwei Li (Academic Editor)

Reviewers' comments:

Reviewer's Responses to Questions

**Comments to the Author**

1. Is the manuscript technically sound, and do the data support the conclusions?

Reviewer #1: Yes

Reviewer #2: Yes

Reviewer #3: Partly

2. Has the statistical analysis been performed appropriately and rigorously? 

Reviewer #1: Yes

Reviewer #2: Yes

Reviewer #3: No

3. Have the authors made all data underlying the findings in their manuscript fully available?

Reviewer #1: Yes

Reviewer #2: Yes

Reviewer #3: No

4. Is the manuscript presented in an intelligible fashion and written in standard English?

Reviewer #1: Yes

Reviewer #2: Yes

Reviewer #3: No

5. Review Comments to the Author

Reviewer #1: I have carefully reviewed the manuscript entitled: "Deformation stage division based on the surface behavior characteristics and internal rupture characteristics of toppling failure kinematic features". The manuscript attempts to establish a relationship between slope surface characteristics and internal rupture features, aiming to determine the internal rock mass rupture characteristics from surface features and explore the deformation stages of anti-dip rocky slopes. This study holds significant innovative value, particularly for the classification of engineering slope evolution stages and the assessment of engineering slope stability. However, due to issues with language, figures, and writing logic, the manuscript requires substantial revisions to improve its clarity and rigor. Below are my detailed comments:

Major points that must be addressed:

1- Language and Readability: I would suggest the authors have it checked preferably by a native English-speaking person to avoid any mistakes. Also, the manuscript should be revised for technical terms and typographical errors. Lastly, ensure consistent use of terms. (For example, in line 17, the abstract should be written as a single paragraph; the current formatting is incorrect. There is inconsistency in the use of technical terms, such as "anti-dip stratified rock slope" in line 12, "anti-dip layered rock slopes" in line 18, and "anaclinal slopes" in line 48. The terms "toppling" and "tippling" are also used inconsistently throughout the manuscript. Additionally, lines 31-36 contain repetitive content with differing references? There are numerous instances of inconsistent terminology and repeated content in the manuscript. Please review carefully.)

2-Line 99: Why was the toe of the slope model cut off? What is the basis for this decision?

3-Lines 101-104: I did not find information regarding the prototype slope in the manuscript. How were the geometric similarity ratio and the physical-mechanical parameters determined?

4-Line 128-140: In the analysis of experimental phenomena, the manuscript describes changes in the maximum and minimum principal stresses at the slope toe. However, this phenomenon is not clearly evidenced in the model experiments. Please provide an explanation for this.

5-Lines 208-249: The authors describe the relationship between internal fracture characteristics and surface deformation, but most of the relationships remain at a qualitative level. Are there more quantitative analyses to determine the correlation between internal deformation and surface characteristics?

6-Line 251: The section "Evolution trend of deformation indicators" is missing, please add it.

7-Lines 424-467: The validation of field data against model results should be elaborated. Explain how field observations support the model predictions and discuss any discrepancies.

Minor points that must be addressed:

8-Figure: Images should be self-explanatory, yet most images in this manuscript lack legends. I recommend the authors add legends to all images. Additionally, some images are unclear and excessive in number. It is suggested that the authors improve image clarity and consolidate similar images.

9-Line 25: Keywords serve to encapsulate the manuscript's content, enabling readers to swiftly identify and understand the topic and scope. In this manuscript, the current keywords do not adequately reflect the research subject. I suggest the authors revise them accordingly.

10-Line 173: I do not understand the phrase 'which is situated in the best air condition.' Does this refer to the 'free face'?

11-Lines 289-291: Regarding interlayer forces, many scholars have conducted research. In fact, the distribution of interlayer forces varies across different stages. Is it appropriate to assume a triangular linear distribution of interlayer forces in this context?

12-The literature review is required to be updated. Authors can use the following papers in the literature review (https://doi.org/10.1007/s12583-023-1835-1 ; https://doi.org/10.1016/j.enggeo.2023.107340 ; https://doi.org/10.1016/j.gsf.2022.101493 ; https://doi.org/10.1007/s11440-022-01627-0 ).

Reviewer #2: This study establishes a coupling relationship between the slope topple deformation patterns and surface macro-deformation characteristics through centrifuge model test, analyzing slope bending and topple deformation stages. The reviewer thinks the topic discussed in this paper is very important, which is of great significance for research of landslide. This reviewer sees that a minor revision will be needed before being accepted for possible publication. Here are the main comments for the revision.

(1) This paper lacks a detailed introduction to actual landslides. In the centrifugal model test, the similarity ratio was 1:100. What is the size of the model box? What is the actual size of the landslide? Is there a review of the similarity ratio relationship?

(2) The introduction section lacks some important references, such as:

10.1007/s10064-021-02135-3; 10.16285/j.rsm.2016.2518; 10.1016/j.soildyn.2019.105961.

(3) Many of the figures in the paper have inconsistent fonts, and some of them are not clear (Fig.2-8, 16, 20…).

(4) Some professional vocabulary is not used properly, for example, "front edge" is suggested to be changed to "toe".

(5) The author was suggested to supplement the sources, basic parameters, image time, and interpretation methods used for SAR images.

(6) In addition to the deformation law, what is the relationship between the deformation amount and deformation rate of landslides obtained from model experiments and Insar data?

(7) The author is suggested to add some textual descriptions in section 3.4.

(8) The cumulative displacement thresholds of landslides lack detailed theoretical basis. Suggest supplementing relevant references.

(9) The conclusion is not concise and innovative. I believe that the Authors should try to interpret and explain more clearly their results. Some key quantitative conclusions should be supplemented.

(10) In this study, the authors must improve the statements about what is new in their study and what are the contributions to the developments of landslide with toppling failure kinematic features.

Reviewer #3: Overall Assessment:

Reject

General Overview:

The manuscript aims at analyzing the failure process of the toppling failure within the anti-dip rock slope, and establishing the relationship between surface behavior and internal fracturing characteristics. The author has conducted extensive research and achieved some meaningful results. Nevertheless, I found some aspects of this paper that lead me to recommend its rejection:

1. The title needs to be rewritten.

2. When an author employs an abbreviation, the full term should appear at the first instance of the abbreviation.

3. “tipping” is suggested to be “toppling”, please check the whole manuscript

4. Insert a space between the unit and the number.

5. Introduction- In fact, the research on the correlation between macroscopic deformation characteristics and internal fractures of anti-dip slopes has been quite comprehensive. On one hand, it is suggested that the authors include references from the past three years concerning studies on toppling failures. On the other hand, the innovative aspects of this paper need further enhancement to emphasize the originality and significance of this research.

6. The manuscript has too many sections, please consolidate some sections to make the structure of the article more concise. For example, Sections 2.1 to 2.4 can be merged into one section.

7. Line 91 Wrong expression “Table 3Ⅰ-a” should be “Figure 3I-a”

8. Line 98 “Observation points were equidistantly placed on the surface” Where are observation points located? Please mark the locations on the figure.

9. Section 2.3. Why the geometric similarity ratio is set of 1:100? What is the prototype rock type of the similar materials? Does the strength of the similar material meet the requirements of prototype rock? Please provide additional details in the manuscript.

10. Line 130 “the max deviator stress（1−3）near the foot of the slope is the largest” . Please supplement the stress monitoring data to support this statement.

11. Lines 133-134 “While the small principal stress drops to zero or even changes to tensile stress, the maximum principal stress hits the highest point near the bottom of the slope.” Please supplement the stress monitoring data to support this statement.

12. Line 146 Spelling mistake “slop” should be “slope”.

13. Line 158. “Figure 8” should be “Fig. 7”. In addition, the expression of the figure should be standardized according to the journal's requirements.

14. Line 183 “No. 4” should be “No. 5”.

15. Lines 234-235 “and tension cracks began to form at a degree of stress of about 1°.” I do not understand this sentence, especially the expression “degree of stress of about 1°”. More explanation needed to be added.

16. The content in Section 3.4 is missing.

17. Line 309, “According to the previous centrifugal test, combined with Huang Runqiu’s deformation system”. Which research? What is Huang Runqiu’s deformation system? Please add related references and statement.

18. Line 318. “it is also necessary to subtract 5 cm” Why is 5 cm? It appears that the author intends to correlate the theoretical derivation results with the model test results. However, there seems to be a lack of scientific basis for this approach.

19. Line 322 “Goodman and Bray proposed that the overturning deformation…”. Please add related references.

20. Line 345 The author did not label Zone B in Fig. 12.

21. Fig. 17. What does the term” deformation intensity” mean? In addition, please add a legend to each contour map in Fig. 17.

22. Please add a scale bar to Fig. 21.

23. Line 457, The Zha Yong toppling deformation is in the III stage of slope dumping deformation but without explanation, please supplement. It is recommended that the authors supplement field photographs of slope deformation and compare them with physical model experiments to validate the accuracy of the conclusions. Currently, the classification of toppling evolution stages appears overly subjective, lacking quantitative criteria for assessment and support from field evidence.

24. Linking InSAR with the division of landslide evolutionary stages is a very interesting idea. However, in previous studies, the formation of large-scale toppling deformations in southwest China has been proven to occur during the valley evolution process. In another words, the toppling period is extremely long. Even if you obtain data for a period using InSAR, it is still very short relative to the existence time of toppling.

25. Language- in additon to the terminology issues, there are still a lot of problems regarding the English usage and the expression. The language at places is imprecise. The problems start from the first sentence of Abstract.

6. PLOS authors have the option to publish the peer review history of their article (what does this mean?). If published, this will include your full peer review and any attached files.

Reviewer #1: No

Reviewer #2: No

Reviewer #3: No

---

## [Author Response · Author response to Decision Letter 0]

31 Aug 2024

Response to Reviewers

Journal Requirements:

Answer: Dear Editor, thank you for the reminder, I have revised the formatting of my thesis manuscript

Answer: Dear Editor, Thank you for the reminder that I have touched up the manuscript of my thesis.

3. Thank you for stating the following in the Acknowledgments Section of your manuscript: "This study is funded by the National Key R&D Program of China (No. 2021YFC3000401)."Please note that funding information should not appear in the Acknowledgments section or other areas of your manuscript. We will only publish funding information present in the Funding Statement section of the online submission form. Please remove any funding-related text from the manuscript. 

Answer: I have removed the acknowledgments.

4. Thank you for stating the following financial disclosure: "This study is funded by the National Key R&D Program of China (No. 2021YFC3000401)." Please state what role the funders took in the study. If the funders had no role, please state: "The funders had no role in study design, data collection and analysis, decision to publish, or preparation of the manuscript." If this statement is not correct you must amend it as needed. Please include this amended Role of Funder statement in your cover letter; we will change the online submission form on your behalf.

Answer: The funders had no role in study design, data collection and analysis, decision to publish, or preparation of the manuscript.

5. We note that your Data Availability Statement is currently as follows: “All relevant data are within the manuscript and its Supporting Information files. “Please confirm at this time whether or not your submission contains all raw data required to replicate the results of your study. Authors must share the “minimal data set” for their submission.

Answer: We're sure that we provided all relevant data, and put the data in supporting information.

Answer: Dear editor, I have applied for ORCID.

7. Please ensure that you refer to Figures 1, and 3 in your text as, if accepted, production will need this reference to link the reader to the figure.

Answer: We accept.

8. 8. We note that Figures 18, 19, and 21 in your submission contain satellite images which may be copyrighted. All PLOS content is published under the Creative Commons Attribution License (CC BY 4.0), which means that the manuscript, images, and Supporting Information files will be freely available online, and any third party is permitted to access, download, copy, distribute, and use these materials in any way, even commercially, with proper attribution. For these reasons, we cannot publish previously copyrighted maps or satellite images created using proprietary data, such as Google software (Google Maps, Street View, and Earth). For more information, see our copyright guidelines: http://journals.plos.org/plosone/s/licenses-and-copyright.

SGS National Map Viewer (https://www.usgs.gov/tools/national-map-viewer)

USGS Earth Resources Observatory and Science (EROS) Center (https://www.usgs.gov/centers/eros)

The Gateway to Astronaut Photography of Earth (https://eol.jsc.nasa.gov/)

Maps at the CIA (https://www.cia.gov/the-world-factbook/maps/)

NASA Earth Observatory (https://earthobservatory.nasa.gov/)

Landsat (https://landsat.visibleearth.nasa.gov/)

Natural Earth (https://www.naturalearthdata.com/)

Answer: I have modified the original Fig. 18 to Fig. 2. Since none of the websites you provided provide a map of the prototype of the landslide studied in this article, this article selects the drone orthophoto of the field survey on the drawing of Figure 2. I deleted Figure 19 from the original manuscript. The original Figure 21 is Figure 20 of the new manuscript, using the C-band Sentinel-1A radar satellite imagery launched by ESA for the wide-area surface deformation detection satellite in the study area, and the archived data is free for all users. Figure 21 in the new manuscript is a photograph taken with a fieldwork camera and a drone

Additional Editor Comments: 

Reviewer #1:

1. Language and Readability: I would suggest the authors have it checked preferably by a native English-speaking person to avoid any mistakes. Also, the manuscript should be revised for technical terms and typographical errors. Lastly, ensure consistent use of terms.

Answer: Dear reviewer, we have revised the article on the issues you raised and touched up the article.

2. Line 99: Why was the toe of the slope model cut off? What is the basis for this decision?

Answer: The slope prototype is located in the Jinsha River basin, and cutting the foot of the modeled slope simulates the erosion of the foot of the slope prototype due to the Jinsha River scour.

3. Lines 101-104: I did not find information regarding the prototype slope in the manuscript. How were the geometric similarity ratio and the physical-mechanical parameters determined?

Answer: Thanks for your kind reminders. We have added the information of the prototype slope, taking into account the capacity of the centrifuge, the size of the model box and the maximum centrifugal acceleration, the maximum centrifugal acceleration used in the test was finally determined to be 100g.

4. Line 128-140: In the analysis of experimental phenomena, the manuscript describes changes in the maximum and minimum principal stresses at the slope toe. However, this phenomenon is not clearly evidenced in the model experiments. Please provide an explanation for this.

Answer：Thanks for your kind reminders. According to Principles of Engineering Geological Analysis and Engineering Geology: Rock in Engineering Construction, during slope formation, the rock mass around the critical surface undergoes unloading and rebound, causing stress redistribution and stress concentration effects. Under the influence of stress differentiation, a stress concentration zone is generated near the critical face. The maximum principal stress near the foot of the slope increases and the minimum principal stress decreases, so that the difference between the maximum and minimum principal stresses here is the largest, which makes the rock body at the foot of the slope the most easily to be cracked and produce damage. Combining the previous studies with experimental phenomena, Fig. 9 and Fig. 19 shows that the foot of the slope is the first place where stress concentration occurs, and the first place to be damaged is the foot of the slope.

5. Lines 208-249: The authors describe the relationship between internal fracture characteristics and surface deformation, but most of the relationships remain at a qualitative level. Are there more quantitative analyses to determine the correlation between internal deformation and surface characteristics?

Answer：Dear reviewer, thank you for your question. In this paper, the qualitative analysis is carried out in the newly revised Chapter 4 (original lines 208-249) when describing the relationship between internal fracture characteristics and surface deformation, but in the newly revised Chapter 2, the relationship between internal fracture characteristics and surface deformation is quantitatively analysed in four aspects, namely, rate cloud map, cumulative displacement change characteristics, point displacement at each point of the 5 slab, and rate change at the point of the #1#, respectively, and at the same time, Fig. 12 demonstrates the internal and surface fracture development pattern of the slope.

6. Line 251: The section "Evolution trend of deformation indicators" is missing, please add it.

Answer: Thanks for your kind reminders. We have modified the order of the images by deleting section 3.4 and placing figures 7-10 from the section in the previous text. What was once Fig.7 is now Fig.5, what was once Fig.8 is now Fig.10, what was once Fig.9 is now Fig.6, and what was once Fig.10 is now Fig.12.

7. Lines 424-467: The validation of field data against model results should be elaborated. Explain how field observations support the model predictions and discuss any discrepancies.

Answer: Thanks for your kind reminders. We have made revisions and added relevant pictures of the site investigation and described them in the text to secondly improve the accuracy of the conclusions

8. Figure: Images should be self-explanatory, yet most images in this manuscript lack legends. I recommend the authors add legends to all images. Additionally, some images are unclear and excessive in number. It is suggested that the authors improve image clarity and consolidate similar images.

Answer: Thanks for your kind reminders, pictures have been modified

9. Line 25: Keywords serve to encapsulate the manuscript's content, enabling readers to swiftly identify and understand the topic and scope. In this manuscript, the current keywords do not adequately reflect the research subject. I suggest the authors revise them accordingly.

Answer: Thanks for your kind reminders. We changed the keywords to anti-dip stratiform rocky slope; centrifuge model test; early identification; macroscopic deformation characteristics

10. I do not understand the phrase 'which is situated in the best air condition.' Does this refer to the 'free face'?

Answer: Yes, we have changed my expression from 'the best air condition' to 'the free face'.

11. Lines 289-291: Regarding interlayer forces, many scholars have conducted research. In fact, the distribution of interlayer forces varies across different stages. Is it appropriate to assume a triangular linear distribution of interlayer forces in this context?

Answer: In this paper, it is proposed to "assume that the gravity loads act on the cantilever slabs in the form of triangular linear distribution of loads", and it is reasonable to assume the triangular linear distribution of interlayer forces for the following reasons: 1.Rock Layer Properties: The physical parameters (e.g., strength, deformability) of the rock layers in the centrifuge experiment do not vary significantly with depth. 2.Stress Distribution Theory: According to soil mechanics principles, the vertical stress distribution in a slope or foundation typically takes a triangular form. For an anti-dip slope, the interlayer force distribution, which represents the shear stress within the slope, can also be assumed to follow a triangular distribution. 3.Slope Geometry and Boundary Conditions: The geometry of the slope constructed in the centrifuge model is relatively regular, and the boundary conditions promote the interlayer force distribution to tend towards a linear triangular form, which is a reasonable assumption. 4.Centrifugal Loading: In the centrifuge model test, the primary load acting on the slope is the centrifugal force, which is relatively stable and does not exhibit pronounced dynamic effects. This allows the influence of the load to be simplified as a triangular linear distribution.5. Initial Deformation Stages: During the initial deformation stages (Stages I and II) of the slope, the deformation of the rock slabs is relatively small, and the interlayer force distribution remains close to the triangular linear assumption. This justifies the use of the "triangular linear distribution of gravity load on the cantilever slabs" to obtain the "ultimate displacement Umax" in these stages. 6.Slope Deformation Limit: As long as the cumulative deformation of the slope does not reach the limit value of deformation and instability (Umax), the slope will not progress into the more advanced stages of damage, even if it encounters various working conditions during its geological history. This further supports the reasonableness of the triangular linear distribution assumption for the interlayer force in the Stages I and II.

12. The literature review is required to be updated. Authors can use the following papers in the literature review.

(https://doi.org/10.1007/s12583-023-1835-1;https://doi.org/10.1016/j.enggeo.2023.107340;https://doi.org/10.1016/j.gsf.2022.101493 ; https://doi.org/10.1007/s11440-022-01627-0 ).

Answer: Sincerest thanks, which we have carefully read and cited. (LINE 47-51,68-74,59-66)

Reviewer #2:

1. This paper lacks a detailed introduction to actual landslides. In the centrifugal model test, the similarity ratio was 1:100. What is the size of the model box? What is the actual size of the landslide? Is there a review of the similarity ratio relationship?

Answer: Thanks for your kind reminders. We've added the size of the model box: length x width x height = 100 cm x 60 cm x 100 cm. The actual size of the landslide in the text: length 2949m, width 2496m.

2. The introduction section lacks some important references

(10.1007/s10064-021-02135-3;10.16285/j.rsm.2016.2518;10.1016/j.soildyn.2019.105961)

Answer: Thank you for recommending the literature, but this article is about loess landslides, so we reselected a 2024, a 2023 published literature on anti-dip stratiform rocky slope, read it carefully, and cited it. (LINE 81-87) 

3. Many of the figures in the paper have inconsistent fonts, and some of them are not clear (Fig.2-8, 16, 20…).

Answer: Thanks for your kind reminders. We've rechecked all the images and revised it.

4. Some professional vocabulary is not used properly, for example, "front edge" is suggested to be changed to "toe".

Answer: Thanks for your kind reminders. We've changed my expression to make it more accurate and professional.

5. The author was suggested to supplement the sources, basic parameters, image time, and interpretation methods used for SAR images.

Answer: Thanks for pointing our question out, we've added the InSAR information to the article.

6. In addition to the deformation law, what is the relationship between the deformation amount and deformation rate of landslides obtained from model experiments and Insar data?

Answer: Dear reviewers, I have the following answers to this question. The denser interferometric fringes and the larger colour difference in the Stacking-InSAR stacked phase map results mean that the deformation phase here is larger. According to the interpretation results of InSAR, in the process of Stacking-InSAR interferometric image identification, the effect of down-track identification is more obvious, the deformation is mostly concentrated in the steep and slow junction, and the deformation degree is greater near the top of the slope. The annual average deformation rate is greater. Since this paper is to study the relationship between the deformation pattern of the anticlinal laminated rocky slope and the macro deformation characteristics and did not carry out a detailed deformation rate interpretation of the InSAR data, only the deformation pattern of the Zayong deformed body was interpreted using the InSAR data to correspond with the deformation pattern of the modelling experiments, to confirm the feasibility of the study.

7. The author is suggested to add some textual descriptions in section 3.4.

Answer: Thanks for your kind reminders. We have modified the order of the images by deleting section 3.4 and placing figures 7-10 from the section in the previous text. What was once Fig.7 being now Fig.3, what was once Fig.8 being now Fig.10, what was once Fig.9 being now Fig.4, and what was once Fig.10 being now Fig.5.

8. The cumulative displacement thresholds of landslides lack detai

---

## [Decision Letter · Decision Letter 1]

22 Sep 2024

PONE-D-24-28956R1Research on the relationship between the overturning deformation pattern and macro-deformation characteristics of anti-dip stratiform rocky slopesPLOS ONE

Dear Dr. Zhang,

Thank you for submitting your manuscript to PLOS ONE. After careful consideration, we feel that it has merit but does not fully meet PLOS ONE’s publication criteria as it currently stands. Therefore, we invite you to submit a revised version of the manuscript that addresses the points raised during the review process.

**ACADEMIC EDITOR: **

The 2rd review process is now complete, and we have two reports submitted by the expert reviewers.

I am very pleased to inform you that after 1st round of modifications, two reviewers both request revisions before the manuscript is accepted. However, the authors must be aware that the comments of reviewers #3 who suggested rejection in the 1st review process are still serious now. 

Therefore, I suggest the authors carefully revise this manuscript point by point according to all the comments. In the end, I want to thank the reviewers for their great efforts on your manuscript and you for your submission.

All the best,

Dr. Linwei Li 

Academic Editor

PLOS ONE

We look forward to receiving your revised manuscript.

Kind regards,

Linwei Li

Academic Editor

PLOS ONE

Additional Editor Comments:

NaN

Reviewers' comments:

Reviewer's Responses to Questions

**Comments to the Author**

1. If the authors have adequately addressed your comments raised in a previous round of review and you feel that this manuscript is now acceptable for publication, you may indicate that here to bypass the “Comments to the Author” section, enter your conflict of interest statement in the “Confidential to Editor” section, and submit your "Accept" recommendation.

Reviewer #1: All comments have been addressed

Reviewer #3: (No Response)

2. Is the manuscript technically sound, and do the data support the conclusions?

Reviewer #1: Yes

Reviewer #3: Yes

3. Has the statistical analysis been performed appropriately and rigorously? 

Reviewer #1: Yes

Reviewer #3: Yes

4. Have the authors made all data underlying the findings in their manuscript fully available?

Reviewer #1: Yes

Reviewer #3: Yes

5. Is the manuscript presented in an intelligible fashion and written in standard English?

Reviewer #1: Yes

Reviewer #3: No

6. Review Comments to the Author

Reviewer #1: The authors have carefully revised the manuscript and addressed all the issues raised by me. It is an interesting work. This version can be accepted for publication after two minor suggestions

1 The title might be better "Relationship between the overturning deformation pattern and macro-deformation characteristics of anti-dip stratiform rocky slopes" or "overturning deformation pattern and macro-deformation characteristics of anti-dip stratiform rocky slopes"

2 Summarized protocol for the physical model test is suggested (as shown in Table 10 at https://doi.org/10.1016/j.gsf.2022.101493)

Reviewer #3: 1. The title still needs to be revised. I recommend revising the title to: ” Relationship between the surface and internal deformation features of toppling in anti-dip rock slopes”.

2. Please ensure consistency in the terminology throughout the entire text. For example:” dump” is suggested to be replaced by “topple”.

3. I did not see any observation points in Fig. 14.

4. Fig.8d- 11d. Do the figures of “rate cloud diagram” refer to horizontal velocity, vertical velocity or total velocity? Is it the average velocity at that stage or instantaneous velocity? More details should be added.

5. From Fig. 6, it is evident that the slope has already experienced failure when the acceleration reaches 48.9 g. However, as per design specifications, the stress field of the model can only match the actual conditions when the acceleration reaches 100 g. In such a scenario, can the results of the model tests truly represent the actual conditions?

6. Chapter 4, the logic is not sufficiently clear, and there is repetition among sections. The author can revise this chapter based on the following suggestions:

4.1 Evolution process of the toppling failure: Divide the toppling evolution stages based on the velocity field, deformation field, and the kinematic characteristics of rock columns, clearly outlining the criteria for stage division and the deformation characteristics of slopes in each stage;

4.2 Relationship between surface and internal deformation features: Focus on analyzing the surface and internal deformation characteristics at different evolution stages and their relationships.

7. Line 185 “Fig. 6Ⅱ-a”, maybe “ Fig. 11Ⅱ-a”.

8. I recommend that the authors provide some practical case studies in the study area to validate the accuracy and practicality of the proposed methods in section 5.2.

9. What does the “distance” in Fig. 17 represent?

10. I did not see any legends in Fig. 19.

11. It is advisable to consolidate Chapters 4 and 6.

12. I recommend changing the title of Chapter 7 to "Case Study”. In this chapter, the authors can validate the correspondence between surface and subsurface deformation characteristics from the previous test and the accuracy of the proposed evaluation methods in chapter 5.

13. Language- Partly improved. But there are still some problems regarding the English usage. Below are just few examples. However, the entire text, including figure captions, has to be carefully edited.

(1) Lines 111 “ deformers” bad expression.

(2) Lines 112 ”slope body” bad expression.

7. PLOS authors have the option to publish the peer review history of their article (what does this mean?). If published, this will include your full peer review and any attached files.

Reviewer #1: No

Reviewer #3: No

---

## [Author Response · Author response to Decision Letter 1]

25 Sep 2024

Response to Reviewers

Additional Editor Comments: 

Reviewer #1:

 The title might be better "Relationship between the overturning deformation pattern and macro-deformation characteristics of anti-dip stratiform rocky slopes" or "overturning deformation pattern and macro-deformation characteristics of anti-dip stratiform rocky slopes"

Answer：Thank you very much for your comments, and after my reflection, I changed the title of my paper from“Research on the relationship between the overturning deformation pattern and macro-deformation characteristics of anti-dip stratiform rocky slopes ”to“Relationship between the surface and internal deformation features of toppling in anti-dip rock slopes”

 Summarized protocol for the physical model test is suggested (as shown in Table 10 at https://doi.org/10.1016/j.gsf.2022.101493)

Answer：Dear Reviewer,I have carefully studied the table 10 you provided for my paper, but I don't believe this paper is suitable for such a detailed description. The reasons are as follows: 1. This paper does not have that many parameters that need to be described. 2.The experiment only involved the application of gravity as a factor.3. Putting the experimental model parameters in a table format would not be able to intuitively demonstrate the construction of the experimental model.

Reviewer #3:

 The title still needs to be revised. I recommend revising the title to:” Relationship between the surface and internal deformation features of toppling in anti-dip rock slopes”.

Answer：Thank you very much for your comments, and after my reflection, I changed the title of my paper from“Research on the relationship between the overturning deformation pattern and macro-deformation characteristics of anti-dip stratiform rocky slopes ”to“Relationship between the surface and internal deformation features of toppling in anti-dip rock slopes”

 Please ensure consistency in the terminology throughout the entire text. For example:” dump” is suggested to be replaced by “topple”.

Answer：Sorry, I apologize for my mistake, I've replaced all the “dump”with “topple”and checked the full text.

 I did not see any observation points in Fig. 14.

Answer：I apologize, but during the subsequent revision of the article, I changed the order of the figures, so the reference here should be to Figure 16, with the observation points being the black and white circular cross markers in that figure. As there are many observation points, directly depicting them in Figure 4(a) in the article would not be aesthetically pleasing. To facilitate your review, I have depicted them more clearly in Figure 16 and have also added a photograph of the actual model setup for comparison.

Fig.16

Actual piling photo

 Fig.8d- 11d. Do the figures of “rate cloud diagram” refer to horizontal velocity, vertical velocity or total velocity? Is it the average velocity at that stage or instantaneous velocity? More details should be added.

Answer: The numbers in the rate cloud diagram refer to the average rate of total displacement change within a given stage. I have incorporated this clarification into the article text.

 From Fig. 6, it is evident that the slope has already experienced failure when the acceleration reaches 48.9 g. However, as per design specifications, the stress field of the model can only match the actual conditions when the acceleration reaches 100 g. In such a scenario, can the results of the model tests truly represent the actual conditions?

Answer: Dear Reviewer, I have carefully considered the question you raised, and I will explain the reasons for the slope failure when the acceleration reaches 48.9g in the following text.

In actual rock masses, there are many cracks and joints, and the rock layer thickness is often not very thick, usually less than 50 cm. If the on-site rock layer thickness is scaled down at a ratio of 1:100, the experimental specimen would only be a few millimeters thick. During the model fabrication process, specimen boards with a thickness below 0.5 cm are very fragile and difficult to make successfully. In order to conduct the experiment, I increased the specimen thickness to 1 cm. In fact, even making 1 cm thick specimens was extremely challenging.

Moreover, the actual slope height is very high. Therefore, when constructing the model, we increased the thickness of the specimens. To correctly reflect the overall strength of the jointed and fractured rock mass, we appropriately reduced the strength of the individual specimens during the fabrication process.

The differential equations of flexural deformation for the rock plate in the model and the prototype are respectively:

 （1.1）

 （1.2）

EIrepresents the flexural rigidity of the rock plate; E represents the elastic modulus of the rock plate; I represents the moment of inertia of the cross-section of the rock plate. For a rectangular cross-section rock plate, I=(bh^3)/12, where *b* is the width of the rectangular cross-section under bending, and ℎ is the thickness (height) of the rectangular cross-section in the bending direction.

According to the similarity theory (scaling law), the similarity ratios between the model (subscript m) and the prototype (subscript p) for the elastic modulus E, cross-sectional moment of inertia I, geometric dimension l, density ρ, and acceleration a are respectivel

S_E=E_p/E_m ，S_I=I_p/I_m ，S_l=l_p/l_m ，S_ρ=ρ_p/ρ_m ，S_a=a_p/a_m 

Substituting the above similarity ratios into Equation (1.2), it can be transformed as:

 （1.3）

Since Equation (1.2) and Equation (1.3) are both valid, the coefficients of the corresponding terms must be equal:

 （1.4）

In the centrifuge model test, *Sρ*=1，*Sa*=1/*n*，*S*l=*n*，therefore，S_E S_I=n^4

That is, when the model and the prototype are similar, the flexural rigidity (EI) of the rock plate in the model and the prototype should satisfy the following relationship:

 （1.5）

In this experiment, the parameters of the model rock plate are Em=2800MPa, bm =10cm, hm =1cm. The prototype rock mass is moderately to slightly weathered phyllite, with an elastic modulus of Ep=17.2GPa and geometric dimensions of bp=1000cm, hp =10cm.

Substituting these values into Equation (1.5), we get n≈49.8.

The bending and toppling mode not only needs to consider the geometric dimensions and the compressive and tensile strengths of the rock mass, but also the flexural rigidity. It is difficult to balance these three factors, as improving one aspect often compromises the other two. In this paper, the focus was on the geometric dimensions and the compressive and tensile strengths of the rock mass, while the flexural rigidity problem was overlooked. In fact, based on the above calculations, the flexural rigidity would fail at around 49.8g of centrifugal acceleration.

Considering the above factors, the concentrated deformation failure of the model occurred at around 50g.

Furthermore, the focus of our experiment is to reproduce the bending deformation process of the rock layers under anti-dip conditions, and to divide the deformation stages, so as to use the surface deformation characteristics as evidence to guide the InSAR interpretation. As long as the surface macroscopic deformation features and the rock layer bending deformation stages are consistent with the actual conditions, the results of this model test can be accepted and used.

 Chapter 4, the logic is not sufficiently clear, and there is repetition among sections. The author can revise this chapter based on the following suggestions:

4.1 Evolution process of the toppling failure: Divide the toppling evolution stages based on the velocity field, deformation field, and the kinematic characteristics of rock columns, clearly outlining the criteria for stage division and the deformation characteristics of slopes in each stage;

4.2 Relationship between surface and internal deformation features: Focus on analyzing the surface and internal deformation characteristics at different evolution stages and their relationships.

Answer：Thank you very much for your suggestions. I have made the changes in the text accordingly.

 Line 185 “Fig. 6Ⅱ-a”, maybe “ Fig. 11Ⅱ-a”

Answer：Thank you for pointing out my mistake. I have changed the label from “Fig. 6Ⅱ-a” to “ Fig. 11Ⅱ-a”and apologize for my oversight.

 I recommend that the authors provide some practical case studies in the study area to validate the accuracy and practicality of the proposed methods in section 5.2.

Answer：Thank you very much for your suggestions. Regarding the verification of the method proposed in Section 5.2, this paper compares the results calculated based on U_MAX with the maximum value of the cumulative displacement curve (Figure 7) of point 1 at the time of slope failure, and finds that the two results are similar. This verifies the reasonableness of the U_MAX inference, thereby confirming the accuracy and practicality of the method proposed in Section 5.2.

 What does the “distance” in Fig. 17 represent?

Answer：The distance here represents the distance between the selected point T and the toe point B. For example, when the distance is indicated as 20, it represents the distance between the yellow point and the green point at the slope toe.

Distance Schematic Diagram

 I did not see any legends in Fig. 19

Answer：I'm very sorry, in the previous version I uploaded an old version of the figure. I have now updated figure 19

Fig. 19

 It is advisable to consolidate Chapters 4 and 6.

Answer：Dear Reviewer,Thank you for your valuable suggestions. Chapter 4 focuses on describing the correspondence between the macro-deformation characteristics of the surface layer, the experimental phenomena, and the image analysis of the centrifuge model test data. Chapter 6 summarizes the correspondence between the toppling deformation patterns and the macro-deformation characteristics of the anti-dip rock slopes. Chapter 6 is derived from Chapter 4. Many of the sliding rules of anti-dip rock slopes are summarized from the centrifuge model tests, moving from specific to general. Based on the centrifuge model tests, the development process of key surface cracks is proposed as a monitoring means for the deformation and failure of such slopes. Combining the characteristics of the velocity contour maps, the physical test phenomena, and the relative cumulative displacement of rock plates at different locations, the deformation schematic diagrams for each stage of the toppling deformation are obtained. In summary, I'm afraid I cannot merge Chapters 4 and 6, but I have reduced the content of Chapter 6 to make the expression more concise.

 I recommend changing the title of Chapter 7 to "Case Study”. In this chapter, the authors can validate the correspondence between surface and subsurface deformation characteristics from the previous test and the accuracy of the proposed evaluation methods in chapter 5.

Answer：Thank you for your suggestions. However, the primary purpose of this paper is to establish the coupling relationship between the slope deformation patterns and the surface macroscopic deformation characteristics. By integrating InSAR remote sensing interpretation and actual ground investigation, we can infer the stage of the slope and provide technical support for the early identification and prevention of major landslide disasters. The landslide case study in Chapter 7 is the specific example used for the centrifuge model experiments in this paper. The purpose of writing Chapter 7 is to verify that the research on the toppling deformation patterns of anti-dip rock slopes and their corresponding surface macroscopic deformation characteristics can guide the InSAR remote sensing interpretation. Due to the above reasons, I have not made any changes to the title of Chapter 7.

 Language- Partly improved. But there are still some problems regarding the English usage. Below are just few examples. However, the entire text, including figure captions, has to be carefully edited.

(1) Lines 111 “ deformers” bad expression.

(2) Lines 112 ”slope body” bad expression.

Answer：Thank you for your suggestions. I have re-polished the language in the article.

---

## [Decision Letter · Decision Letter 2]

8 Oct 2024

PONE-D-24-28956R2Relationship between the surface and internal deformation features of toppling in anti-dip rock slopesPLOS ONE

Dear Dr. Zhang,

Thank you for submitting your manuscript to PLOS ONE. After careful consideration, we feel that it has merit but does not fully meet PLOS ONE’s publication criteria as it currently stands. Therefore, we invite you to submit a revised version of the manuscript that addresses the points raised during the review process.

**ACADEMIC EDITOR: **

I have received the referee report for your paper, from which you can see that although the reviewer generally support publication, there are some issues that need to be properly addressed. Therefore, I encourage you to promptly make revisions to address these issues. 

If you have any questions or need further clarification on the revisions required, please do not hesitate to contact me. I am here to assist you throughout this process.

Thank you for your attention to this matter, and I look forward to receiving your revised manuscript.

Meanwhile, I want to thank the reviewers for their great efforts on your manuscript and you for your submission.

All the best,

Dr. Linwei Li 

Academic Editor

PLOS ONE

We look forward to receiving your revised manuscript.

Kind regards,

Linwei Li

Academic Editor

PLOS ONE

**Journal Requirements:**

Reviewers' comments:

Reviewer's Responses to Questions

**Comments to the Author**

1. If the authors have adequately addressed your comments raised in a previous round of review and you feel that this manuscript is now acceptable for publication, you may indicate that here to bypass the “Comments to the Author” section, enter your conflict of interest statement in the “Confidential to Editor” section, and submit your "Accept" recommendation.

Reviewer #3: (No Response)

2. Is the manuscript technically sound, and do the data support the conclusions?

Reviewer #3: Yes

3. Has the statistical analysis been performed appropriately and rigorously? 

Reviewer #3: Yes

4. Have the authors made all data underlying the findings in their manuscript fully available?

Reviewer #3: Yes

5. Is the manuscript presented in an intelligible fashion and written in standard English?

Reviewer #3: No

6. Review Comments to the Author

**Reviewer #3:** The authors attended to most of the comments made by the reviewers. Before the article is accepted, there are still some issues that need to be amended.

1. The legend for the deformation intensity in Figure 19 should be quantified, and there is a lack of units to represent the deformation intensity. In addition, the concept of deformation strength intensity should be explained in the text.

2. The expression of “distance” in Fig. 17 should be added in the text.

3. The language of the manuscript still needs further enhancement.

7. PLOS authors have the option to publish the peer review history of their article (what does this mean?). If published, this will include your full peer review and any attached files.

Reviewer #3: No

---

## [Author Response · Author response to Decision Letter 2]

9 Oct 2024

Response to Reviewers

Additional Editor Comments: 

Reviewer #3:

1. The legend for the deformation intensity in Figure 19 should be quantified, and there is a lack of units to represent the deformation intensity. In addition, the concept of deformation strength intensity should be explained in the text.

Answer：Figure 19 is based on the characteristics of the velocity cloud and the physical test phenomena, combined with the relative cumulative displacement of the rock plates at different stages. It shows the gradual transfer of the main deformation zone from the slope toe to the slope top on the anti-dip stratiform rocky slopes. In fact, Figure 19 is a generalization of the velocity cloud diagram, representing the changing trend of the main deformation zone distribution at different stages of this type of toppling deformation body, as well as the relative deformation severity in each region. The reason why the legend cannot be quantified in this paper is that Figure 19 represents a type of landslide rather than a specific one, so the different colors in the legend only represent the varying degrees of deformation severity, from red to blue indicating a decrease from severe to mild deformation. To better represent the changing trend of the deformation zone distribution, I have changed the title of Figure 19 to "Distribution of the severity of toppling deformation".

2. The expression of “distance” in Fig. 17 should be added in the text.

Answer：Dear Reviewer, thank you for the reminder. I have added the explanation of the meaning of "Distance" in Figure 17 into the main text.

3. The language of the manuscript still needs further enhancement 

Answer：Dear Reviewer, I have carefully re-polished the entire manuscript. I hope it can meet your approval.

---

## [Editor Report · Decision Letter 3]

11 Oct 2024

Relationship between the surface and internal deformation features of toppling in anti-dip rock slopes

PONE-D-24-28956R3

Dear Dr. Zhang,

We’re pleased to inform you that your manuscript has been judged scientifically suitable for publication and will be formally accepted for publication once it meets all outstanding technical requirements.

Kind regards,

Linwei Li

Academic Editor

PLOS ONE

Additional Editor Comments (optional):

NaN
---

## [Editor Report · Acceptance letter]

28 Oct 2024

PONE-D-24-28956R3 

PLOS ONE

Dear Dr. Zhang, 

I'm pleased to inform you that your manuscript has been deemed suitable for publication in PLOS ONE. Congratulations! Your manuscript is now being handed over to our production team.

Kind regards, 

on behalf of

Dr. Linwei Li 

Academic Editor

PLOS ONE